# Neuroendocrine modulation sustains the *C. elegans* forward motor state

Maria A Lim[1]*, Jyothsna Chitturi[1,2], Valeriya Laskova[1,3], Jun Meng[1,3], Daniel Findeis[4], Anne Wiekenberg[4], Ben Mulcahy[1], Linjiao Luo[5], Yan Li[1,3], Yangning Lu[1,3], Wesley Hung[1], Yixin Qu[1], Chi-Yip Ho[1], Douglas Holmyard[1], Ni Ji[6,7], Rebecca McWhirter[8], Aravinthan DT Samuel[6,7], David M Miller[8], Ralf Schnabel[4], John A Calarco[9]*, Mei Zhen[1,2,3]*

[1]Lunenfeld-Tanenbaum Research Institute, Mount Sinai Hospital, Toronto, Canada; [2]Institute of Medical Science, University of Toronto, Toronto, Canada; [3]Department of Physiology, University of Toronto, Toronto, Canada; [4]Institut für Genetik, Technische Universität Braunschweig Carolo Wilhelmina, Braunschweig, Germany; [5]Key Laboratory of Modern Acoustics, Ministry of Education, Department of Physics, Nanjing University, Nanjing, China; [6]Center for Brain Science, Harvard University, Cambridge, United States; [7]Department of Physics, Harvard University, Cambridge, United States; [8]Department of Cell and Developmental Biology, Vanderbilt University, Nashville, United States; [9]FAS Center for Systems Biology, Harvard University, Cambridge, United States

*For correspondence: lim@lunenfeld.ca (MAL); jcalarco@fas.harvard.edu (JAC); zhen@lunenfeld.ca (MZ)

Competing interests: The authors declare that no competing interests exist.

**Abstract** Neuromodulators shape neural circuit dynamics. Combining electron microscopy, genetics, transcriptome profiling, calcium imaging, and optogenetics, we discovered a peptidergic neuron that modulates *C. elegans* motor circuit dynamics. The Six/SO-family homeobox transcription factor UNC-39 governs lineage-specific neurogenesis to give rise to a neuron RID. RID bears the anatomic hallmarks of a specialized endocrine neuron: it harbors near-exclusive dense core vesicles that cluster periodically along the axon, and expresses multiple neuropeptides, including the FMRF-amide-related FLP-14. RID activity increases during forward movement. Ablating RID reduces the sustainability of forward movement, a phenotype partially recapitulated by removing FLP-14. Optogenetic depolarization of RID prolongs forward movement, an effect reduced in the absence of FLP-14. Together, these results establish the role of a neuroendocrine cell RID in sustaining a specific behavioral state in *C. elegans*.

## Introduction

While the hard-wiring of circuits through chemical and electrical synapses provides the anatomic basis for behaviors, their outputs are subjected to modulation and reconfiguration by neuromodulators such as biogenic amines and neuropeptides (reviewed in *Katz, 1998*; *Marder et al., 2014*; *Taghert and Nitabach, 2012*). Unlike classical neurotransmitters that activate receptors at the vicinity of their release sites (active zones) to trigger rapid and short-lived responses, neuromodulators, released by the dense core vesicles (DCVs) either peri-synaptically or volumetrically, may diffuse over long distances and posses longer half-lives (reviewed in *Nässel, 2009*). Consequently, they can evoke long-range and prolonged responses through metabotropic G-protein-coupled receptors.

Neuroendocrine cells and neuromodulators have been examined in various animal models (*Hartenstein, 2006*; *Marder and Bucher, 2007*; *van den Pol, 2012*). Neuromodulation exerts profound and lasting effects on physiology and behaviors across phyla. In many invertebrates,

neuromodulators may be co-released with classic neurotransmitters at the presynaptic termini to exert local effects on circuits, or, they may be secreted from specialized peptidergic neurons that are devoid of active zones to exert long-range or global effects on neural networks (*Nässel, 2009*). Importantly, studies in invertebrate models made key contributions to the principle that neurons and their connections can be modified by a vast network of neuromodulators, enabling hard-wired circuits to generate flexible activity patterns underlying different states and adaptive behaviors.

Various neuromodulators that act on invertebrate circuits to affect movement, aggregation, sleep, arousal, and learning behaviors have been reported (*Frooninckx et al., 2012*; *Marder and Bucher, 2007*; *Taghert and Nitabach, 2012*). One of the best-described examples is the crustacean stomatogastric ganglion (STG), a single rhythm generating circuit that drives different motor patterns of the gut to either grind or filter food (*Harris-Warrick et al., 1992*). A variety of amines and neuropeptides, secreted from the descending and ascending peptidergic neurons directly on the STG neuropils, and likely also from the hemolymph, act on every neuron and synapse in this circuit, reconfiguring the network to produce different output patterns (*Blitz et al., 2008*; *Marder and Bucher, 2007*). Similarly, in the feeding circuit of the sea slug *Aplysia*, the balance between the activity of two interneurons, which determines whether food is ingested or egested, is regulated by neuropeptides (*Jing et al., 2007*; *Taghert and Nitabach, 2012*). In *Drosophila* larvae, local modulatory neurons in the ventral nerve cord secrete biogenic amines to activate and modify crawling patterns (*Selcho et al., 2012*). In vertebrates, the hypothalamus produces and releases many hormones to control appetite, reproduction, circadian rhythms, and social behaviors (*Graebner et al., 2015*; *Schwartz et al., 1996*; *Selcho et al., 2012*; *Strauss and Meyer, 1962*). Similarly, the vertebrate spinal cord locomotory circuits are activated and modified by multiple amines that are secreted by descending neurons at the brainstem, as well as by secretory neurons that reside locally in lower vertebrates such as the lamprey (reviewed in *Miles and Sillar, 2011*).

Recent genomic analyses hinted the evolutionary origin of peptidergic signaling to be the last common bilaterian ancestor (*Mirabeau and Joly, 2013*). Evidence for cross-species functional conservation of certain neuromodulators is also emerging. Indeed, serotonin and dopamine signaling have been known to modify vertebrate and invertebrate motor function alike. Pigment Dispersing Factor (PDF), a neuropeptide that couples the motor rhythm with the circadian clock in *Drosophila* (*Renn et al., 1999*), also couples the motor rhythm with the developmental clock, cyclic molting, in *C. elegans* (*Choi et al., 2013*; *Li and Kim, 2010*; *Raizen et al., 2008*). Oxytocin and vasopressin peptides exert strong effects on sexually dimorphic behaviors (*Benarroch, 2013*), as well as cognition in both mammals and the nematode *C. elegans* (*Beets et al., 2012*; *Garrison et al., 2012*). A better understanding of neurosecretory cell development and function across the animal phyla is required to reveal the origin and principles that underlie neural network architecture and operation.

*C. elegans* allows for a genetic dissection of the roles of neuromodulators on behaviors (*Frooninckx et al., 2012*; *Holden-Dye and Walker, 2013*; *Koelle, 2016*). In addition to biogenic amine neuromodulators, its genome contains 113 genes that may encode up to 250 distinct peptides of three classes (*Li and Kim, 2010*): the insulin-like (INS) (*Pierce et al., 2001*), FMRF-amide-related (FLP), and non-insulin/non-FMRF-amide-related (NLP) (*Husson et al., 2007*). A number of these neuropeptides already have assigned roles, affecting 'simple' (locomotion, feeding, egg-laying) to 'complex' (mating, lethargus, aggregation, learning) behaviors (*Beets et al., 2012*; *Bendena et al., 2008*; *Bhattacharya et al., 2014*; *Chalasani et al., 2007*; *Chen et al., 2013*; *de Bono and Bargmann, 1998*; *Garrison et al., 2012*; *Hums et al., 2016*; *Janssen et al., 2009*; *Macosko et al., 2009*; *Turek et al., 2016*; *Waggoner et al., 2000*; others). For instance, the NLP-type PDF-1 increases velocity and suppresses reversals through premotor and other interneurons (*Flavell et al., 2013*; *Meelkop et al., 2012*). The insulin-like peptide INS-1 is secreted by an interneuron to alter salt taxis behavior upon starvation (*Tomioka et al., 2006*). NPR-1, a receptor for the FLP-type FLP-18 and FLP-21, mediates context-dependent avoidance and aggregation behavior (*Choi et al., 2013*).

The presence and morphology of specialized neuroendocrine systems have been described in many invertebrates and vertebrates (*Hartenstein, 2006*; *van den Pol, 2012*). In comparison, the morphology and identity of the *C. elegans* neuroendocrine system remains to be better described and clearly defined. *Albertson and Thomson (1976)* reported the first candidate *C. elegans* secretory neuron (NSM) to contain both clear and dense core vesicles. *White et al. (1986)* described another neuron (BDU) to 'have striking, darkly staining vesicles'. Reporters for neuropeptide-

encoding genes are found to exhibit expression across the nervous system. Many *C. elegans* modulators of known functions act through neurons that clearly participate in classical chemical synaptic transmission. These notions led to a speculation that all *C. elegans* neurons secrete neuromodulators (*Holden-Dye and Walker, 2013*), which predicts distributed peptidergic signaling.

However, as noted in *White et al. (1986)*, the *C. elegans* samples utilized in these classic anatomic studies were optimized for visualization of active zones, not vesicle or other intracellular organelle preservation. This made anatomic determination of neurons with specialized endocrine properties – devoid of active zones in other systems – challenging. We recently completed the serial electron microscope (EM) reconstruction of several *C. elegans* larvae (Witvliet et al. in preparation), fixed by tannic acid and high-pressure freezing that allows for better preservation of intracellular structures in physiological states (*Rostaing et al., 2004*; *Weimer, 2006*). These and our other unpublished EM studies reveal two distinct sources for neuropeptides. First, all classical chemical synapses contain a small portion of DCVs, typically residing at the periphery of presynaptic termini. Second, a small cohort of dedicated secretory neurons can be distinguished by the prevalence of DCVs, which can be further divided into two subgroups. One, including NSM, BDU, and a few other neurons, exhibits mixed anatomic features that may equally support both secretory and synaptic transmission; the other exhibits predominantly secretory anatomic features. RID, a neuron currently of incomplete anatomic reconstruction (*White et al., 1986*), and of unknown function, belongs to the second group. Here we describe its anatomy, development and function.

The survival of animals depends on their ability to generate appropriate motor responses to a dynamic environment. Like all other animals, *C. elegans* responds to cues from their surroundings by altering motor strategies. How a hard-wired sensorimotor circuit adjusts its output constitutes a central question for neuroscience. Through anatomic, developmental, and functional analyses, we discover that RID is a peptidergic neuron that modulates the *C. elegans* motor state to sustain forward movement, and it does so in part through secreting a FMRF-amide family neuropeptide FLP-14. Our findings lay the groundwork for further interrogation of mechanisms that underlie neuroendocrine development, evolution, and circuit modulation.

## Results

### RID exhibits the ultrastructural hallmarks of a specialized endocrine neuron

We reconstructed the dorsal nerve cord (DNC), a fascicle consisting of multiple motor neuron processes that innervate the dorsal body wall muscles. We reconstructed the entire length of a first stage larva (L1), and fragments of multiple young adults, by serial transmission EM (sTEM) (*Figure 1A–C*; Appendix 1). In all samples, we observed a single process that adopts a stereotypic location in the DNC and exhibits the hallmark features of a specialized neuroendocrine cell. Based on the topology of the fully reconstructed neuron and other criteria (Appendix 1), we identified this process to be the axon of a neuron named RID. As described in *White et al. (1986)*, the RID soma resides in the anterior dorsal head ganglion. It sends a ventral-projecting process that reaches the ventral ganglion, loops around to enter the dorsal ganglion, turns posteriorly to enter the DNC, and runs along the entire length of DNC to reach the tail (illustrated in *Figure 1A*).

Our L1 and adult DNC reconstruction revealed several unreported features for the RID axon and the DNC. The DNC innervates the dorsal body wall muscles through *en passant* neuromuscular junctions (NMJs) (*White et al., 1986*). The RID process resides dorsally within the fascicle, nested between the epidermis and GABAergic (DD) motor neuron processes. It exhibits periodic changes in diameters, from ~40 nm to ~400 nm (in L1s), and ~40 nm to ~750 nm (in adults), creating regularly spaced varicosities at ~1.3 μm (L1) and ~2.5 μm (adult) intervals along its entire length (*Figure 1B, C*). These varicosities intercalate between NMJs. In adults, they typically reside immediately adjacent to NMJs made by the DD motor neurons (*Figure 1C*).

The morphological features of these varicosities are distinct from those of their neighboring NMJs (*Video 1*). In both L1s and adults, NMJs made by all classes of motor neurons contain a mixed vesicle population with a stereotypic structural organization: a large pool of clear, <40 nm synaptic vesicles adjacent to an active zone (415.6 ± 83.2 per bouton, by adult DD), and a small pool of ~40–70 nm DCVs residing at the periphery (10.5 ± 2.4 per bouton, by adult DD) (*Figure 1C*). In contrast,

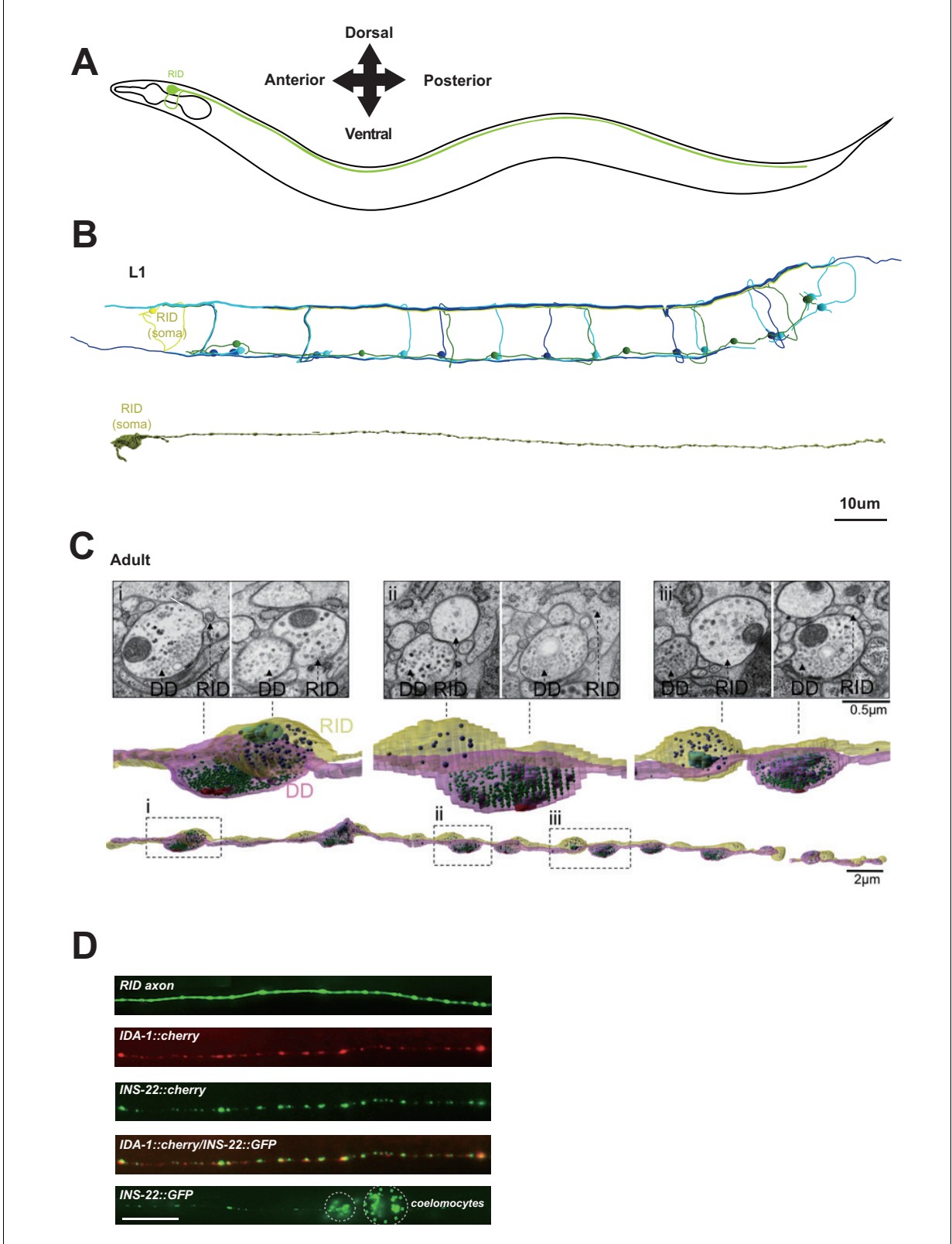

**Figure 1.** RID is a peptidergic neuron. (**A**) Schematic of the RID neuron. (**B**) sTEM reconstruction of RID and motor neurons in a L1 animal. Top panel, Skeletal reconstruction of motor neurons and respective processes in dorsal nerve cord (DNC). Yellow, RID; Green, DD; Light blue, DA; Dark blue, DB. Bottom panel, Volumetric reconstruction of the L1 RID cell body and dorsal cord neurite showed periodic swellings along the RID neurite. (**C**) Volumetric reconstruction of a portion of the RID axon in a young adult. Bottom, The neurite of RID (yellow), DD (pink). Middle, higher magnification

*Figure 1 continued on next page*

*Figure 1 continued*

versions of the regions indicated by the dashed boxes i, ii and iii. Top, Representative EM cross-section images of RID and DD boutons in the adult DNC. In the volumetric reconstruction, green spheres indicate synaptic vesicles (SVs), blue spheres dense core vesicles (DCVs), red shading indicates active zones, and light blue shading indicates mitochondria. (D) Top panels, a cytoplasmic GFP reporter illustrates the RID axon, followed by reporters for the DCV membrane protein IDA-1 and the neuropeptide INS-22 along the RID axon. Bottom panel, INS-22::GFP accumulated at coelomocytes (dotted circle), indicating that it was secreted. Scale bar, 5 µm.

varicosities along the RID axon contain almost exclusively ~40–100 nm DCVs (20.3 ± 7.8 per bouton, by adult RID axon), the vast majority (78 out of 87) devoid of active zones. A small fraction of varicosities (9 out of 87) contain an active zone-like electron density, but none have associated DCVs.

RID can load and release neuropeptides. Consistent with these EM observations, expression of the fluorescently labeled tyrosine phosphatase receptor IDA-1, a DCV marker (*Zahn et al., 2001*), in RID produced a pearling pattern along the DNC, consistent with periodic DCV accumulation at specific locations along its process (*Figure 1D*). The exogenous expression of INS-22::GFP, a *C. elegans* neuropeptide marker (*Sieburth et al., 2005*) in RID resulted in a similar pearling pattern that co-localized with IDA-1::RFP (*Figure 1D*). INS-22::GFP signals also accumulated in coelomocytes (*Figure 1D*), the scavenger cells that endocytose secreted substances. Hence RID exhibits the anatomic features of an endocrine cell that predominantly secretes neuropeptides.

## RID fails to differentiate in the absence of a Six/SO homeobox transcription factor UNC-39

To address the role of RID, we performed a visual screen for genetic mutations that disrupt RID development. Using a fluorescent reporter (*Pceh-10*-GFP) (generated based on (*Altun-Gultekin et al., 2001*; *Svendsen and McGhee, 1995*) that is expressed by four neurons RID, AIY, CAN, ALA and one sheath cell, we isolated a mutant *hp701* that exhibits fully penetrant and specific loss of the marker expression in RID (*Figure 2A*). *hp701* harbors a C to T substitution that results in a recessive and causative missense P298L mutation in *unc-39* (*Figure 2B*; Materials and methods). A canonical loss-of-function *unc-39* allele *e257,* which harbors a R203Q missense mutation (*Yanowitz et al., 2004*), fully recapitulates the RID phenotype of the *hp701* allele (*Figure 2B*) and failed to complement it.

UNC-39 is a homeobox transcription factor of the Six/SO family. Six/SO is necessary for the development of the fruit fly eye (*Cheyette et al., 1994*), the insect endocrine gland corpora cardiac (*De Velasco et al., 2004*), and the vertebrate forebrains (*Lagutin et al., 2003*). *C. elegans unc-39* mutants exhibit variable defects in post-embryonic mesodermal differentiation, soma or neurite migration, and expression of terminal fate markers in several neurons (*Manser and Wood, 1990*; *Yanowitz et al., 2004*). As previously reported (*Yanowitz et al., 2004*), our *unc-39* reporters (Supplemental Methods) were expressed by muscles and multiple neurons during embryonic development and into the adulthood (*Figure 2C*). Notably, we observed robust expression of *unc-39* reporters in the embryonic RID precursor (*Figure 2—figure supplement 1A*), embryonic RID (*Figure 2C*, top panel), and newly hatched L1 larvae (*Figure 2C*, middle panel). Post-embryonically, *unc-39* expression was selectively decreased in some neurons, including RID (*Figure 2C*, lower panels).

Our analyses revealed a complete and specific loss of RID terminal differentiation in both *unc-39* alleles. First, all known fate markers (*kal-1, ser-2, mod-1, unc-3*) for RID (and a few other neurons) (*Bülow et al., 2002*; *Tsalik and Hobert, 2003*; *Wang et al., 2015a*) exhibited a

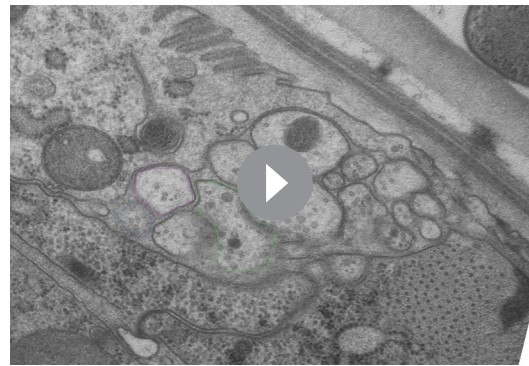

**Video 1.** Thirty-two consecutive serial sections of a part of the dorsal nerve cord in an adult wild-type animal. The RID process is outlined in pink, and the DD axon is outlined in green.

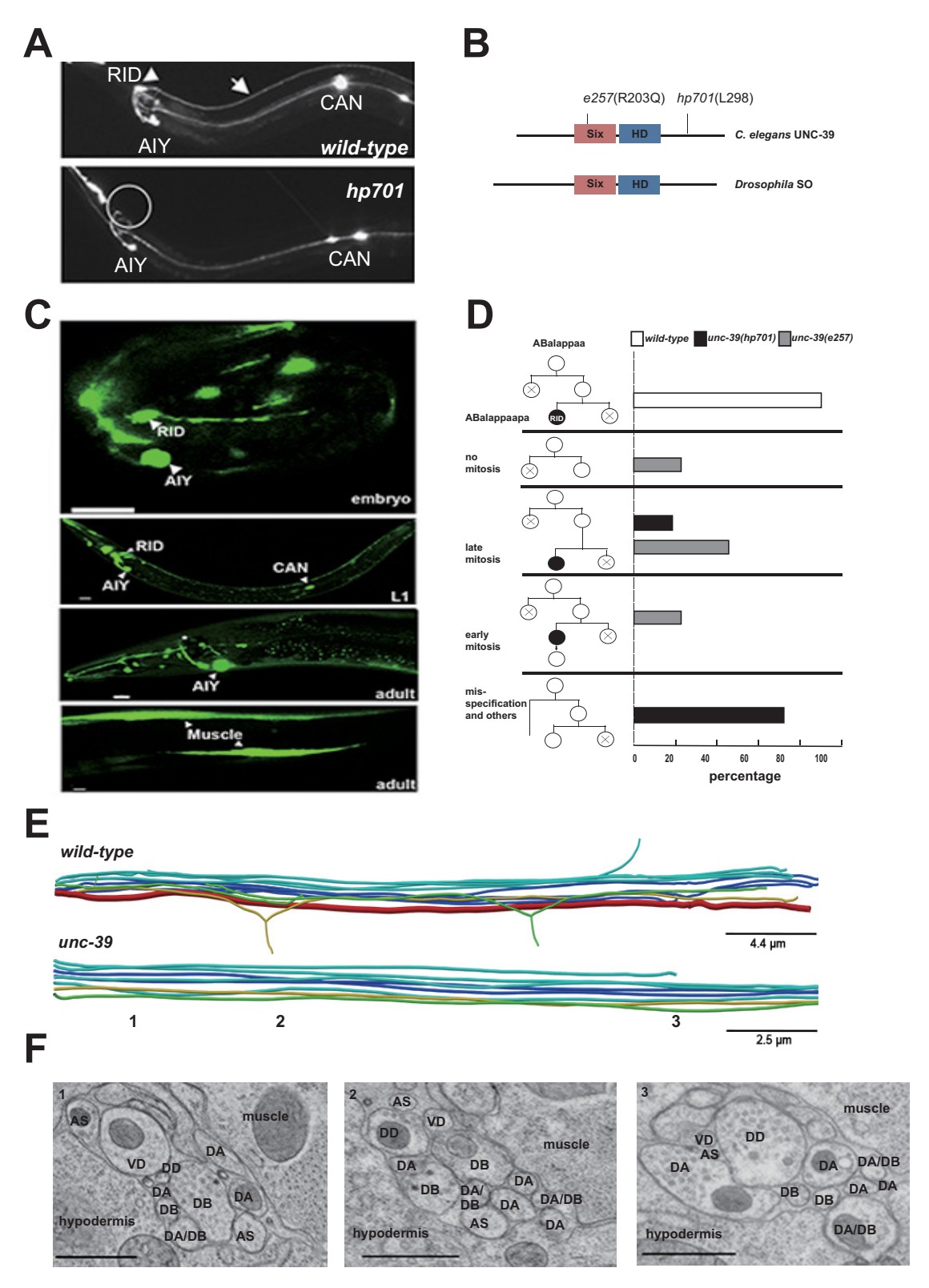

**Figure 2.** RID fails to differentiate in *unc-39* mutants. (**A**) In *unc-39* mutants, RID soma (circle) and axon could not be detected by the *Pceh*-10-GFP marker, while other *Pceh*-10-GFP cells are present. Scale bar, 10 μm. (**B**) A predicted protein structure of UNC-39 compared with its Drosophila homologue SO, denoted with allelic information of *hp701* and *e257* mutations. (**C**) *unc-39* expression in the nervous system was observed in embryos (top panel), newly hatched L1 larvae (middle panel), and young adults (lower two panels). Scale bar, 10 μm. (**D**) Lineage map of ABalappaa, the

*Figure 2 continued on next page*

Figure 2 continued

neuroblast that gives rise to RID (ABalappaapa) in embryos of wild-type animals and *unc-39* mutants. *unc-39* mutants exhibited a range of mitosis changes. (E) Skeletal sTEM reconstruction of a fragment of the DNC of a wild-type and *unc-39* mutant. The RID axon is absent from the *unc-39* DNC. Red, RID; Yellow, DD; Green, VD; Light blue, DA/AS; Dark blue, DB. (F) Representative images of the EM cross-section from the DNC of *unc-39* mutants. Numbers (1-3) denote their approximate locations in E. RID process is absent in all three sections. The identity of neurons, hypodermis and muscles are labeled accordingly. Scale bar, 500 nm.

The following figure supplement is available for figure 2:

**Figure supplement 1.** The expression pattern of UNC-39::GFP and phenotypes of *unc-39* mutants.

complete and specific loss of expression in RID in both *unc-39* alleles (*Figure 2—figure supplement 1B,C*). Second, our sTEM reconstruction of segments of the DNCs of three *unc-39* adults failed to reveal a process that morphologically resembles the RID axon (*Figure 2E*, RID axon denoted in red in wild-type), while all present processes were unambiguously traced back to a motor neuron class (*Figure 2E,F*; Appendix 1). Third, the lineage development that gives rise to RID exhibited fully penetrant mitosis and terminal differentiation defects in *unc-39* embryos. The ABalappaa neuroblast gives rise to RID after two rounds of mitosis, where during each round the sister cells of the RID precursor and RID activate apoptosis (*Sulston et al., 1983*); illustrated in *Figure 2D*). By simultaneous DIC and fluorescent tracing, we confirmed that in wild-type embryos, the progeny of ABalappaa followed stereotypic spatial and temporal patterns for mitosis and apoptosis, and its designated anterior granddaughter turned on the RID (*Pceh-10*) fate marker shortly after the second round of mitosis (*Figure 2D*). In *unc-39* embryos, ABalappaa exhibited a missed, accelerated, or delayed second mitosis; its progeny did not turn on or maintain the *Pceh-10* marker and/or execute apoptosis (*Figure 2D*). Blocking apoptosis in the RID lineage by removing either the lineage-specific (*Wang et al., 2015a*), or global (*Yuan and Horvitz, 1990*) apoptotic activators/executors UNC-3, CED-3 and CED-4 did not restore RID in *unc-39* mutants (*Figure 2—figure supplement 1C,D*), further supporting the notion that the loss of RID is due to failure in differentiation, not ectopic apoptosis.

The loss of RID in *unc-39* mutants exhibits striking lineage-specificity. The CAN, ALA, RMED and RIA neurons are lineage-related to RID by sharing progenitors that give rise to ABalappaa (*Sulston et al., 1983*). They are present in both *unc-39* alleles, with either no or mild and variable defects (e.g. soma position) in adults. RID differentiation thus may be particularly sensitive to the perturbation of UNC-39 activity.

## RID synthesizes multiple neuropeptides, including FLP-14 and INS-17

The morphology of RID suggests that it functions as a specialized peptidergic neuron. To identify neuropeptides that are expressed by RID, we performed mRNA sequencing on differentiated RID neurons isolated from *C. elegans* larvae. Due to the absence of a strong RID-specific promoter that is required for cell sorting, we devised a subtractive transcriptome strategy, taking advantage of the complete and specific loss of RID in *unc-39* larvae to detect RID-enriched transcripts.

Briefly, we FACS-sorted *Pceh-10*-GFP-positive (GFP+) cells from wild-type and *unc-39(hp701)* second stage larvae (*Figure 3A*; Appendix 1). We compared transcripts of wild-type and *unc-39* GFP+ cells against their respective unsorted whole cell populations (All Cells) (*Figure 3B*; Appendix 1) to obtain two enriched transcript datasets for GFP+ cells in wild-type animals (RID, ALA, AIY, CAN, and a sheath cell) and *unc-39* mutants (ALA, AIY, CAN, and a sheath cell). The non-overlapping transcripts thus represent candidates that are either differentially expressed in RID, or differentially regulated by UNC-39 (*Figure 3C*).

In total, we identified 574 enriched transcripts from wild-type animals, and 867 from *unc-39* mutants. 421 transcripts were similarly enriched in both datasets; 599 preferentially enriched in either one of two datasets. Among them, 446 were more enriched in *unc-39* mutant cells, which may represent genes repressed by UNC-39; 153 were more enriched in wild-type cells (*Supplementary file 2*; Appendix 1). The last category represents candidates that are either highly enriched in RID or activated by UNC-39.

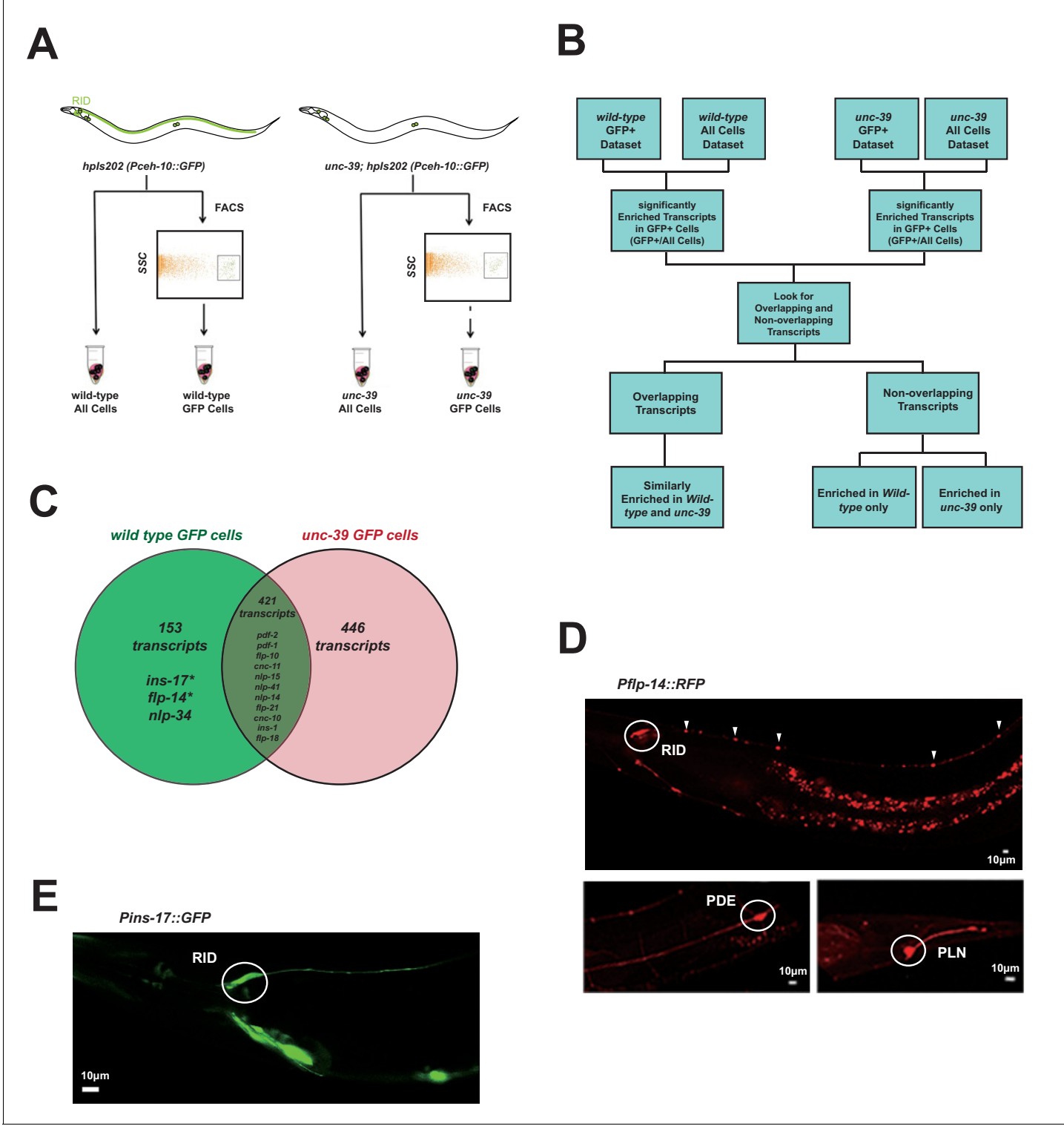

**Figure 3.** Subtractive transcriptome profiling reveals neuropeptides expressed by RID. (**A**) The experimental design and schematic of cell isolation protocol by flow cytometry. SSC, side scatter. (**B**) The workflow of data analysis. (**C**) A venn diagram representation of neuropeptide transcripts enriched in wild-type and *unc-39* datasets. (**D**) A transcriptional reporter of *Pflp-14* exhibits expression in RID cell body (circle), along the RID axon (arrowheads) and other neurons, including those in the mid-body (PDE, bottom left panel) and tail (PLN, bottom right panel). Scale bar, 10 μm. (**E**) A transcriptional reporter of *Pins-17* exhibits expression in RID (circle) and other unidentified neurons. Scale bar, 10 μm.

For the purpose of this study, our follow-up analysis strictly focused on the peptide-encoding transcripts highly enriched in RID. Both wild-type and *unc-39* datasets (*Supplementary file 2*) revealed enrichment for a handful of secreted peptide- or protein-encoding transcripts (*Figure 3C*; *Table 1*; *Supplementary file 2*). As a validation of our approach, transcripts that are similarly enriched in both datasets (*Supplementary file 1*, positive controls) encode peptides expressed by neurons present in both wild-type and *unc-39* larvae, e.g. *nlp-15* (AIY), *hen-1* (CAN, AIY) (*Ishihara et al., 2002*; *Janssen et al., 2009*; *Nathoo et al., 2001*; *Wenick and Hobert, 2004*), and *pdf-1* (ALA; our observation). By contrast, *flp-14*, *ins-17,* and *nlp-34* showed more enrichment in the wild-type dataset, with *flp-14* and *ins-17* exhibiting the most significant difference and abundant expression, making them candidate genes highly-expressed in RID. Reassuringly, using transcriptional reporters (*Figure 3D,E*; Appendix 1), we confirmed that *flp-14* and *ins-17* are robustly expressed in RID and additional neurons. For *flp-14*, these neurons include ALA (weak and variable expression), PDE, and PLN (strong expression, *Figure 3D*). Based on their soma positions, other neurons that consistently express *flp-14* may be AIM, AIY, and AVK.

## RID activity increases during forward locomotion

To address the function of the RID neuron, we monitored its activity pattern in behaving animals by calcium imaging. A genetic calcium sensor GCaMP6, fused with RFP (mCherry), was expressed in RID to monitor relative neuronal activity change. Parameters for spontaneous locomotion (directionality and instantaneous velocity) and neuronal activity (GCaMP/RFP ratio) were simultaneously acquired as described (*Aoyagi et al., 2015*; *Kawano et al., 2011*; *Xie et al., 2013*); Materials and methods).

RID maintained low activity during reversals, and increased its activity during forward movement (*Figure 4A,B*). Increased calcium activity was observed in both the RID soma (*Figure 4*) and axon (*Video 2*). No other *Pflp-14*-positive neurons exhibited obvious activity change that correlated with forward movement. During the transition from reversals to forward movement, we consistently observed an activity rise in RID, typically with a significant lag (~10 s) in reaching its peak when compared to the velocity rise (*Figure 4A*). A decrease in RID activity was observed following the transition from forward movement to reversals (*Figure 4A,B*). There was significant correlation between the rate of velocity change (acceleration and deceleration, X-axis) and the change in RID activity ($Ca^{2+}$ rise and decay, Y-axis), when animals transited between forward movement and reversals, respectively (*Figure 4C*; $p<0.006$ for forward to reversal; $p=0.0007$ for reversal to forward movement).

## RID is required for sustained forward locomotion

The activation of RID during forward movements and the positive correlation between the rate of its activity change with that of velocity change during the forward and reversal transition suggest that RID positively modulates forward movement. To examine this possibility, we quantified the effect of anatomic removal of RID on spontaneous motor behaviors.

**Table 1.** Identification of significantly enriched neuropeptide transcripts in RID by subtractive transcriptome profiling.

| Gene | | Transcript counts (Mean±SD) | | Transcript counts (Mean±SD) | | Enrichment in GFP+ Cells (Fold change) | | P values for the enrichment (FDR-corrected; n >= 3 replica) | |
| --- | --- | --- | --- | --- | --- | --- | --- | --- | --- |
| Class | Sequence ID (Gene Name) | GFP+ cells (wt) | All cells (wt) | GFP+ cells (*unc-39* mutants) | All cells (*unc-39* mutants) | *wild-type* | *unc-39 mutants* | *wild-type* | *unc-39 mutants* |
| Insulin-family peptide | F56F3.6 (*ins-17*) | 265.6 ± 37.8 | 34.4 ± 10.5 | 101.7 ± 68.7 | 56.0 ± 17.8 | 7.7 | 1.8 | 2.37E-06 | 0.72 |
| FLP-family peptide | Y37D8A.15 (*flp-14*) | 19547.2 ± 586.8 | 2917.5 ± 368.0 | 8562.1 ± 6291.9 | 3427.3 ± 1069.2 | 6.7 | 2.5 | 7.55E-08 | 0.53 |
| NLP-family peptide | B0213.17 (*nlp-34*) | 67.0 ± 23.2 | 9.8 ± 6.5 | 37.8 ± 35.3 | 13.3 ± 6.5 | 6.8 | 2.8 | 0.01 | 0.67 |

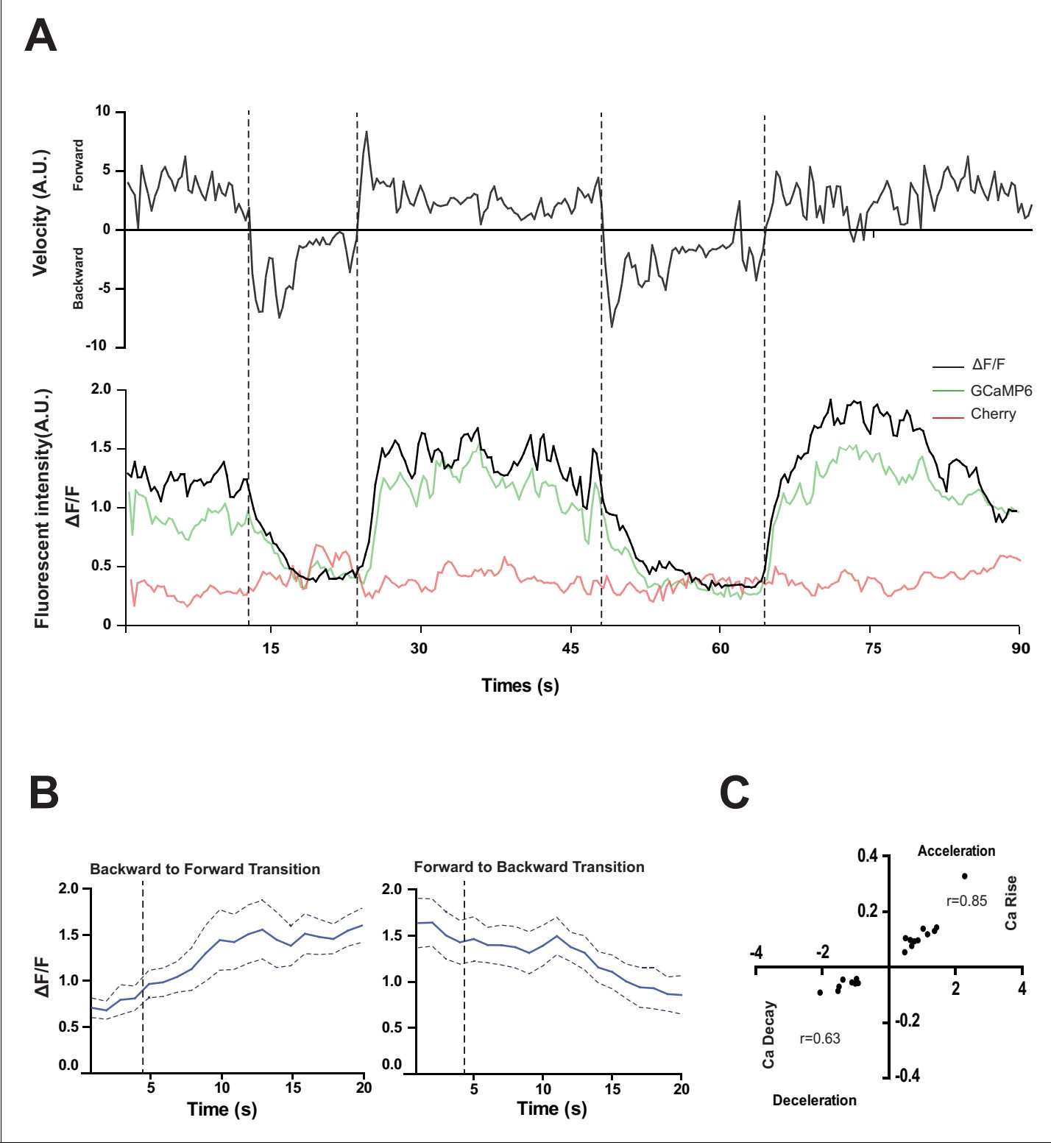

**Figure 4.** RID activity increase correlates with forward movements. (**A**) Representative velocity (top) and corresponding RID calcium activity trace (bottom) from a freely moving animal. Normalized ratiometric signal changes (ΔF/F), as well as the raw fluorescence intensities of GCaMP and cherry are shown. ΔF/F was used to calculate changes in calcium activity for each animal. Changes in positions of fluorescent signals were used to calculate velocity and directionality. (**B**) RID activity as measured by GCaMP/cherry ratio change (± SEM) during transition periods. Left panel, RID activity increased when animals transition from backward to fast forward locomotion. Right panel, RID activity decreased when animals transition from forward

*Figure 4 continued on next page*

*Figure 4 continued*

to backward locomotion. For **A** and **B**, dotted longitudinal lines indicate transition periods from backward to forward locomotion and vice versa. (**C**) Cross-correlation analyses between the change in RID activity and the change in velocity. Positive and negative slopes (Y-axis) indicate increase (Ca rise) and decrease (Ca decay) in RID, respectively. Positive and negative values on the X-axis indicate changes in velocity from backward to forward locomotion (acceleration) and from forward to backward locomotion (deceleration), respectively. For **B** and **C**, N = 10 animals/genotype. In **C**, each dot represents a transitional event.

*C. elegans* executes forward movement that is periodically disrupted by pauses, reversals, and turns (*Gray et al., 2005*; *Pierce-Shimomura et al., 1999*). Our spontaneous motor behavior analyses were performed when wild-type animals exhibit exploratory behaviors (*Gao et al., 2015*; *Kawano et al., 2011*). We quantified the directionality (forward, reversal, pause), sustainability (duration and frequency of re-initiation of movement from pauses), and speed (instantaneous velocity). Under our assay conditions (Appendix 1), wild-type *C. elegans* exhibited a predominant preference for forward movement (>95% of total time) over reversals and pauses (<5% combined). RID-ablated animals drastically reduced the propensity for forward movement (~60% of total time), and increased the propensity for reversals (~20%) and pauses (~20%) (*Figure 5A*). RID-ablated animals exhibited a significant reduction in the duration (*Figure 5A''*) and velocity (*Figure 5A'''*) of each forward run, with more frequent re-initiation of forward movement (*Figure 5A'*). They also increased reversal re-initiation frequency (*Figure 5A'*), without significantly altering reversal duration (*Figure 5A''*). Hence, upon RID ablation, sustained forward movement was replaced by more frequent, shorter and slower forward runs, more pauses, and more reversals.

*unc-39* mutants, harboring an anatomic loss of RID among other defects, exhibited motor characteristics strikingly reminiscent to the RID-ablated animals – replacing long foraging with more frequent, shorter and slower forward runs, more pauses, and more reversals (*Figure 5A–A'''*; *Videos 3* and *4*). These behavioral results complement the RID calcium imaging results, implying not only an association, but also causality of RID activation for sustained foraging.

## *flp-14* mutants partially recapitulate the motor defects caused by RID removal

We next addressed whether the role of RID in sustaining forward runs requires neuropeptides. We first examined the motor behaviors of *flp-14* and *ins-17* null mutants. The *flp-14* locus encodes three identical copies of the KHEYLRF amide peptide. The *flp-14* deletion allele, which removes the entire coding region of the FLP-14 peptide, exhibited motor characteristics that resembled *unc-39* and RID-ablated mutants, but with reduced severity (*Figure 5A–A'''*). *ins-17* mutants, by contrast, exhibited motor behaviors indistinguishable from wild-type animals; *ins-17 flp-14* double mutants showed behavioral defects that mostly resembled *flp-14* (*Figure 5B–B'''*; *Figure 5—figure supplement 1*).

Like RID-ablated or *unc-39* animals, *flp-14* mutants replaced long foraging with shortened forward runs and more frequent pauses and reversals (*Figure 5A–A''*; *Figure 5—figure supplement 1*; *Video 5*). *flp-14* mutants exhibited a reduced severity in their defects: the reduction of mean duration of forward runs, and the increase of re-initiation frequency of forward and reversal movements were less prominent as the RID-ablated or *unc-39* mutant animals. The mean

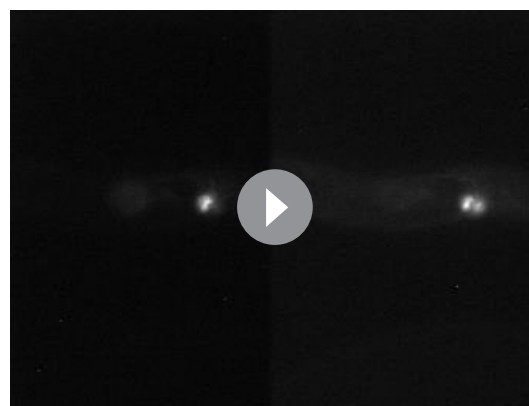

**Video 2.** Changes in the RID calcium transients in moving animals. RID activity increased during a period of acceleration in a forward bout, or during transitions from reversal to fast forward locomotion. Left panel: RFP; Right panel: GCaMP6. Note that multiple neurons expressed GCaMP6::RFP, but only RID (soma and axon) exhibited signal increase that was correlated with increased forward locomotion.

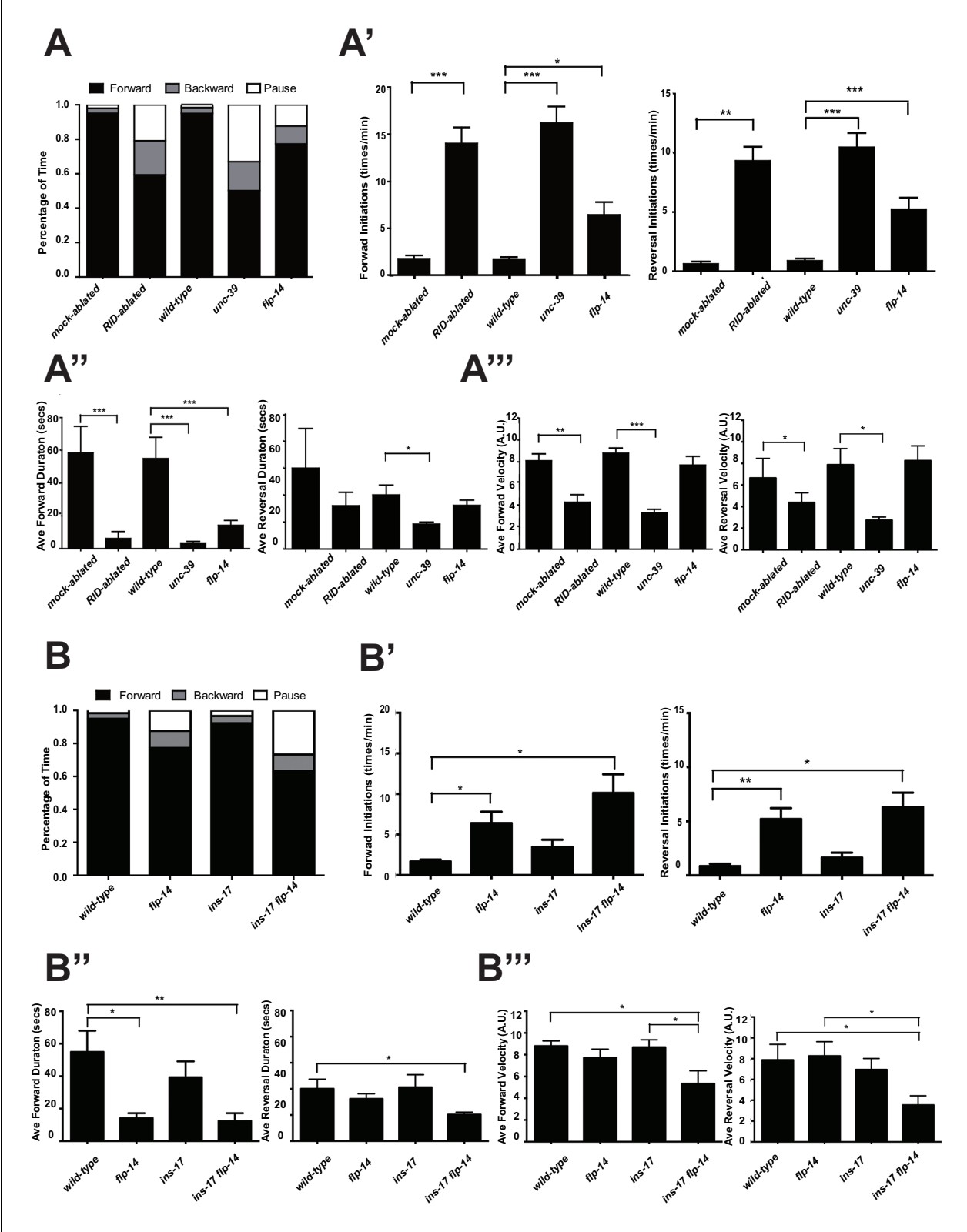

**Figure 5.** RID and FLP-14 potentiate sustained, long forward movements. (**A-A"'**) Spontaneous motor behavioral output, the propensity of directional movement and the interruption of forward movement between wild-type control (mock-ablated *Pceh-10*-GFP *animals*), RID-ablated *Pceh-10*-GFP animals, wild-type (N2), *unc-39*, and *flp-14* mutants. RID-ablated, *unc-39*, and *flp-14* mutants showed decreased propensity for long forward runs, replacing it with more frequent reversals and pauses. (**A**) Total fractional time animals of each genotype spent in forward, reversal, or pauses. (**A'**)
*Figure 5 continued on next page*

*Figure 5 continued*

Frequency of re-initiation of forward runs and reversals. (A'') Duration of forward and reversal runs. (A''') Velocity of forward and reversal runs. (B-B''') Spontaneous motor behavioral output, the propensity of directional movement and the interruption of forward movement between wild-type (N2), *ins-17*, and *ins-17 flp-14* mutants. (B) Total fractional time animals of each genotype spent in forward, reversal, or pauses. (B') Frequency of re-initiation of forward runs and reversals. (B'') Duration of forward and reversal runs. (B''') Velocity of forward and reversal runs. *ins-17* mutants did not show significant changes in motor behavior, while *ins-17 flp-14* generally resembled *flp-14* in motor behavior. N = 10 animals/genotype. Error bars are ± SEM.

The following figure supplements are available for figure 5:

**Figure supplement 1.** Raw data for spontaneous motor behaviors of animals quantified in *Figure 5*.

**Figure supplement 2.** Frequency distribution of forward and reversal velocities quantified in *Figure 5*.

forward and reversal velocity, reduced in both RID ablated and *unc-39* mutant animals, did not exhibit a statistically significant change in *flp-14* mutants (*Figure 5A–A'''*; *Figure 5—figure supplements 1* and *2*). Expressing a single copy of the wild-type *flp-14* genomic fragment *Si(FLP-14)* in *flp-14* mutants led to an increase in the duration of the forward runs and a decrease of the re-initiation frequency for both forward and reversal movements (*Figure 6A–A''', B–B'''*; *Figure 6—figure supplements 1* and *2*). The rescue of these motor defects confirms the functional requirement of FLP-14 in sustaining forward locomotion.

To determine if FLP-14 from RID contributes to its requirement for sustained forward movement, we removed RID either genetically using the *unc-39* mutation (*Figure 6A*; *Figure 6—figure supplement 1*), or by laser ablation (*Figure 6—figure supplement 2*) in *flp-14* animals carrying the rescuing transgene *Si(FLP-14)*. Both manipulations led to reduced duration and velocity in forward runs, and increased bout initiation frequency (*Figure 6*; *Figure 6—figure supplements 1* and *2*). Together, these results suggest that FLP-14 promotes forward movement, and RID is the key cellular origin for such a role. The less severe defects of *flp-14* mutants also implicate the presence of additional effectors through which RID promotes forward runs.

## RID stimulation promotes forward movement, through FLP-14 and other effectors

We further examined whether the role of RID in sustaining the forward motor state involves FLP-14 by examining the effect of optogenetic activation of RID. Through repurposing an endogenous *C. elegans* embryonic ubiquitin-ligase system (*Armenti et al., 2014*), we restricted the expression of Chrimson, a light-activated cation channel (*Klapoetke et al., 2014*) to RID (*Figure 7—figure supplement 1*; Appendix 1). Briefly, we tagged Chrimson with ZF1, a degron recognized by an embryonic-specific E3 ligase ZIF-1, so that selective chrimson degradation in non-RID neurons could be induced by an exogenous expression of ZIF-1 (Details described in Appendix 1). Due to potential concerns on a delay or ineffectiveness in optogenetic stimulation of DCV exocytosis (*Arrigoni and Saper, 2014*), we used a robust stimulation protocol (a three-minute light ON/three minute light OFF cycle; Materials and methods; Appendix 1), where we compared the run length with or without stimulation (*Figure 7A*), as well as the change of mean velocity before and after stimulation during a forward run (*Figure 7B,C*).

Wild-type animals exhibited longer forward runs during periods of stimulation (*Figure 7A*). This effect was abolished in both *flp-14* and *unc-*

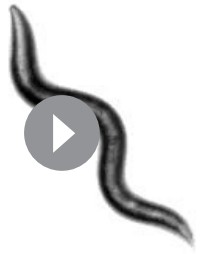

**Video 3.** Representative video of a L4 stage wild-type (N2) animal on an NGM plate with a thin-layer of OP50 bacteria food. Head is at the top at the beginning of the video.

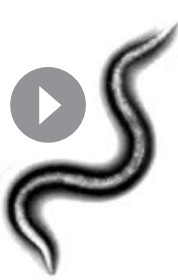

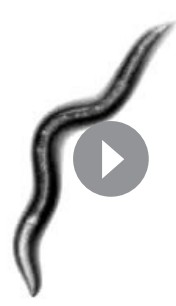

**Video 4.** Representative video of a L4 stage *unc-39 (hp701)* animal on an NGM plate with a thin-layer of OP50 bacteria food. Head is at the bottom at the beginning of the video.

**Video 5.** Representative video of a L4 stage *flp-14 (gk1055)* animal on an NGM plate with a thin-layer of OP50 bacteria food. Head is at the bottom at the beginning of the video.

*39* mutant animals (*Figure 7B,C*). Moreover, when wild-type animals were executing forward movement, stimulation of RID induced a velocity increase (*Figure 7B,C*; *Figure 7—figure supplement 1*; *Video 6*). This effect was reduced and abolished in *flp-14* and *unc-39* mutants, respectively (*Figure 7B,C*; *Figure 7—figure supplement 1*).

Consistent with the observation that RID activity rise and decay significantly lag behind the directional transition, the optogenetic stimulation of RID when animals were executing reversals did not lead to a prompt switch to foraging (*Video 7*). Taken together, all results consistently point to a modulatory, instead of a deterministic role of RID for the motor state: RID is dispensable for *C. elegans* to execute or transit between foraging and reversals, but its activation sustains longer forward runs, and such a role requires a neuropeptide FLP-14 and additional contribution from unidentified factors.

## The loss of two predicted FLP-14 receptors, NPR-4 and NPR-11, does not recapitulate the effect of RID loss

Identifying the physiological receptors of FLP-14 is necessary for determining the downstream signaling of the RID neuron. NPR-11 and NPR-4, two GPCRs expressed by several interneurons (*Chalasani et al., 2007*; *Cohen et al., 2009*) were the predicted candidate FLP-14 receptors (*Frooninckx et al., 2012*; *Holden-Dye and Walker, 2013*). However *npr-4 npr-11* mutant animals exhibited no or very modest motor defects (*Figure 8A*). Optogenetic activation of RID in *npr-4 npr-11* double mutants yielded effects identical to wild-type animals (*Figure 8B–D*; *Figure 7—figure supplement 1*). These results suggest that either the prediction for receptors is incorrect, or, because both NPR-4 and NPR-11 were predicted to bind multiple ligands, their effects on the forward run are obscured by the activity of other ligands and neurons.

## Discussion

Motor circuits must allow output flexibility for animals to select and adjust their motor strategies. Here we identify a previously unknown endocrine component of the *C. elegans* motor circuit, and reveal its role in promoting the forward state. The circuit mechanism that enables *C. elegans* to modulate the propensity of directionality incorporates a specialized peptidergic neuron.

### Similar ultrastructure of invertebrate neuroendocrine cells

The modulatory nature of neural networks is ubiquitous, and likely of ancient evolutionary origin (*Mirabeau and Joly, 2013*; *Scharrer, 1987*). The EM samples used in this study allowed us to characterize the distribution of neuromodulatory cells in the *C. elegans* nervous system. Similar to the case of the crustacean STG (*Kilman and Marder, 1996*), and in the brain (*Maley, 1990*), we

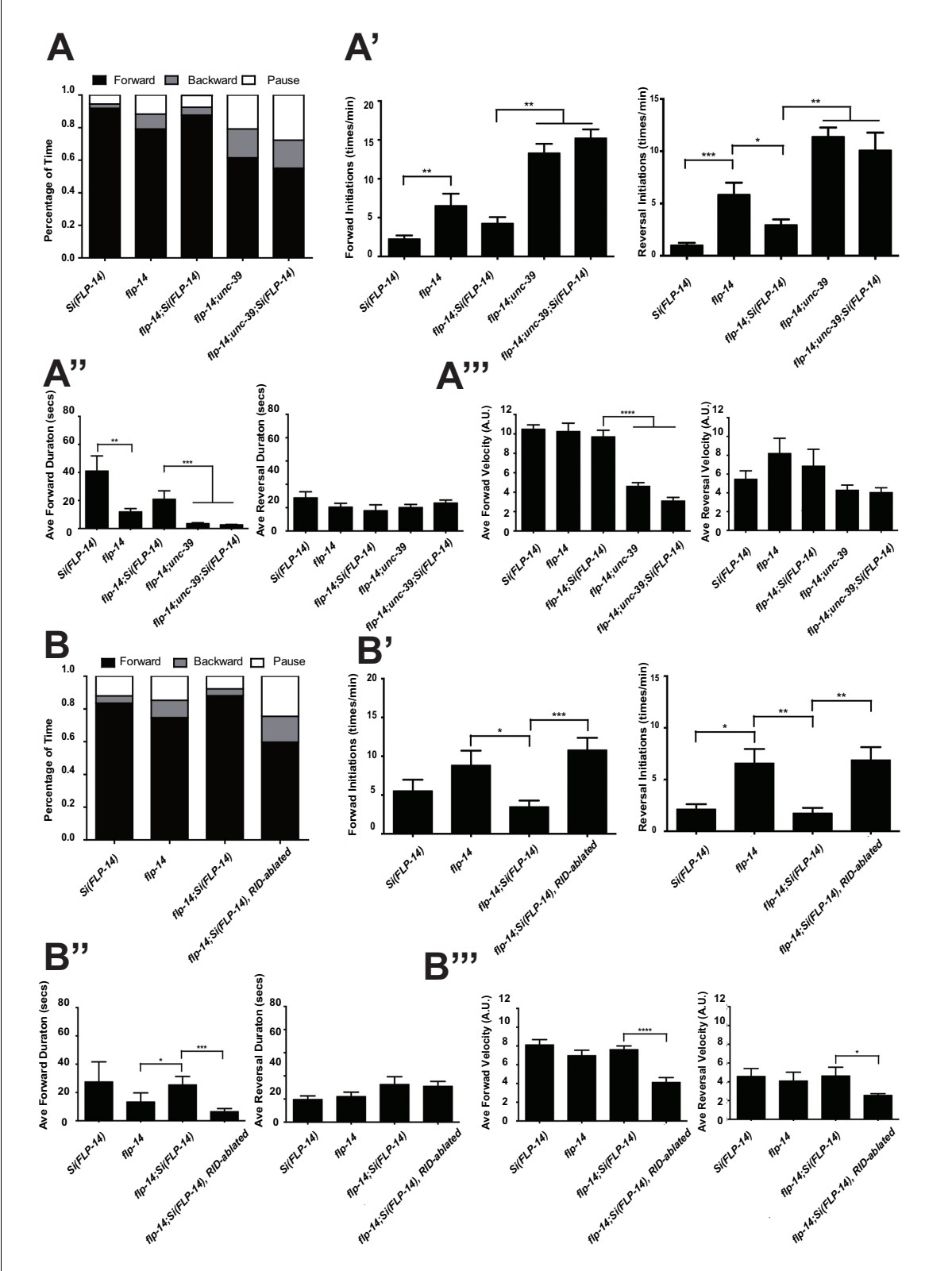

**Figure 6.** FLP-14 potentiates forward movements through RID. (A–A''') Spontaneous motor behavioral output between wild-type *Si(FLP-14)*, *flp-14*, *flp-14;Si (FLP-14)*, and *flp-14;Si (FLP-14);unc-39* where RID was genetically ablated. We quantified the propensity of directional movement (A), the continuity of forward movements, by the re-initiation frequency of forward and backward movement (A') and the duration of forward and backward movement (A''), as well as the mean velocity of forward and backward movement. (B–B''') Spontaneous motor behavioral output between wild-type *Si(FLP-14)*, *flp-*

*Figure 6 continued on next page*

*Figure 6 continued*

*14, flp-14;Si (FLP-14), and flp-14;Si (FLP-14)* where RID was ablated by a laser beam. All animals were in the background of the RID marker transgene (*Pceh-10-GFP*) for this set of experiments. A single copy of FLP-14 reversed *flp-14* mutants' motor defects; this effect was abolished when RID was laser ablated. N = 10 animals/genotype. Error bars are ± SEM.
The following figure supplements are available for figure 6:

**Figure supplement 1.** Raw data for spontaneous motor behaviors of animals quantified in *Figure 6A*.
**Figure supplement 2.** Raw data for spontaneous motor behaviors of animals quantified in *Figure 6B*.

observed a wide distribution of DCVs across most neuropils with different distribution patterns. Most neurons have sparse DCVs localized to the periphery of classic presynaptic termini. But some neurons, such as RID, are specialized endocrine cells that predominantly harbor DCV pools. A precise measurement of the DCV size in this dataset was difficult due to the section thickness (70 nm), but they are easily distinguishable from clear synaptic vesicles by size (>40 nm) and dark appearance. Like synaptic vesicles, they cluster in bouton-like compartments that lack active zones, or, do not tether around them when active zone-like structures are present in the compartment. Hence, contents of DCVs may be co-released at synapses with neurotransmitters, as well as be released diffusely and asynchronously. The ultra-structural criteria for neuroendocrine cells in *C. elegans* are consistent with those in other systems.

## Conservation of RID's peptidergic function among nematodes

*Ascaris suum*, a large parasitic nematode, encodes AF2, a KHEYLRF amide neuropeptide identical to the *C. elegans* FLP-14. By in situ hybridization and mass spectrometry, AF2 was found to be strongly expressed by *Ascaris suum*'s RID-equivalent neuron (*Jarecki et al., 2010*), indicating a conservation of not only overall nervous system organization, but also the cellular and molecular properties between nematode species.

*C. elegans* RID promotes the forward motor state in part through secreting FLP-14. These results are complementary to the reports that a bath application of AF2 to the *Ascaris suum* neuromuscular preparation potentiated the depolarization of body wall muscles and the forward-driving cholinergic motor neurons (*Cowden and Stretton, 1993*; *Pang et al., 1995*; *Trailovic et al., 2005*; *Verma et al., 2007*). These results implicate that RID may also activate body wall muscles and/or forward-driving motor neurons to promote forward movement.

## A multi-layered neuromodulation at motor circuits

Descending interneurons that release neuromodulators to modify the activity pattern of motor circuits have been identified in flies, crustaceans, fish and rodents (reviewed in *Miles and Sillar, 2011*). Being the sole neuron that extends a process throughout the entire length of the dorsal nerve cord, RID resembles a descending peptidergic modulator of the motor circuit. We show that the loss of RID does not abolish the animal's ability to execute forward movement, but reduces its duration and velocity, whilst increasing the propensity for the opposing motor modes. RID exhibits an activity increase after animals switch from reversals to foraging, and a decrease when they transit to reversals. Optogenetic stimulation of RID prolonged forward runs, but was insufficient to cause an immediate switch from the reversal mode to foraging. Notably, RID was one of the neurons that exhibited a coordinated activity pattern with known components of the forward motor circuit in a pan-neuronal imaging study on immobilized animals (*Kato et al., 2015*). All results are consistent with RID being a positive modulator for the forward-driving motor circuit.

Activity of the crustacean STG is influenced by nearly 20 neuromodulators, at each neuron and throughout its connectivity (*Marder and Bucher, 2007*). Similarly, the *C. elegans* motor output can be modulated at multiple layers, by different neuromodulators and from different neuronal origins. Some neuromodulators modify the activity of the core motor circuit components. An escape behavior is facilitated by tyramine released by an interneuron (RIM), which inhibits a premotor interneuron (AVB) to halt forward movement, and simultaneously activates the GABAergic motor neurons (VDs)

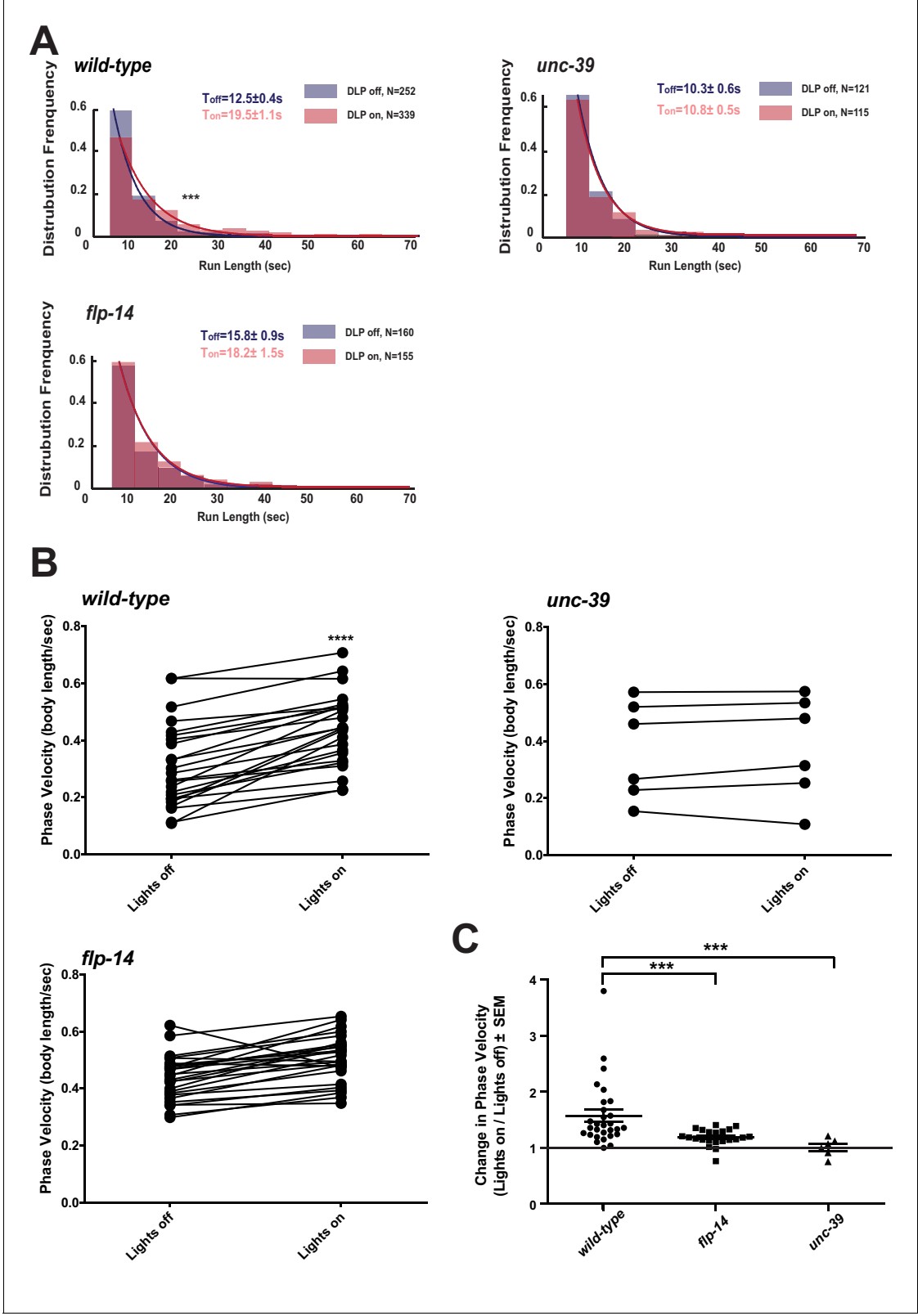

**Figure 7.** Activation of RID promotes forward movements in part through FLP-14. (**A**) The distribution of the mean run length for all light ON (RID stimulation) and light OFF (no RID stimulation) periods. (**B–C**) A comparison of the motor behavior response before and after RID optogenetic stimulation in wild-type, *unc-39*, and *flp-14* animals. The change of speed (phasic velocity) before and after RID stimulation in wild-type, *unc-39* and *flp-*

*Figure 7 continued*

14 animals, respectively, was quantified in **C**. In response to RID stimulation, wild-type animals showed increased velocity and run length during a forward run. This response was abolished in *unc-39* mutants and reduced in *flp-14* mutants. N = 6–26 animals/per genotype.

The following figure supplement is available for figure 7:

**Figure supplement 1.** Restricting chrimson expression in RID by repurposing an embryonic E3 ligase.

to induce deep bending (*Donnelly et al., 2013*; *Pirri et al., 2009*). Upon exposure to food, dopaminergic sensory neurons activate an NLP-12-releasing interneuron (DVA), which acts upon cholinergic motor neurons (*Hu et al., 2011*) to reduce forward runs, retaining animals on food (*Bhattacharya et al., 2014*). Other neuromodulators function through complex and not yet deciphered circuit pathways: PDF-1 promotes forward runs (*Janssen et al., 2009*) and roaming (*Flavell et al., 2013*). It is released by a forward-promoting premotor interneuron (AVB) and other neurons, and may act through multiple and upper layer interneurons (*Flavell et al., 2013*). A NLP-12-releasing interneuron (DVA) coordinates with a FLP-1-releasing interneuron (AVK) to facilitate a body posture change that is associated with halting the forward runs when animals encounter an oxygen reduction (*Hums et al., 2016*). Our results add another layer to motor circuit modulation - a descending input to the motor circuit from a FLP-14-releasing neuron RID.

## The circuit mechanism of RID remains unknown

The *C. elegans* core motor circuit components for executing forward locomotion have been well established: the premotor interneurons (AVB, PVC) provide electrical and chemical synaptic inputs, respectively, to potentiate a cholinergic motor neuron group (the B motor neurons), which innervate and organize muscle contractions in a sequential order to propel the animal forward (reviewed in *Zhen and Samuel, 2015*). How this core circuit activity is regulated is less clear.

A full elucidation of the functional connectivity of RID awaits future investigation (*Figure 8E*), as our current exploration indicates complexity. Two main reported inputs to RID in the adult wiring diagram are the premotor interneurons of the forward motor circuit, PVC and AVB, (*White et al., 1986*). Such a circuit disposition would allow RID to function downstream of the forward-driving premotor interneurons to promote forward bouts. Ablation of PVC and AVB separately (and with other interneurons); however, did not abolish the RID's activity rise during, or coordination with, forward movement (*Figure 8—figure supplement 1A,B,E*). They may act redundantly, or additional inputs may activate RID. A full EM reconstruction of RID in adults is required.

The *C. elegans* adult wiring diagram deduced GABAergic motor neurons and hypodermis as RID's postsynaptic partners, based on their apposition to a few active zones of the RID axon (*White et al., 1986*). Because the majority of RID presynaptic termini do not contain active zones, the peptidergic nature of RID emphasizes the necessity to identify FLP-14 downstream signaling through its receptors, instead of strictly based on the active zone structures. The striking analogy between the anatomic structure and cellular contents of RID in different nematode species suggests that RID may directly activate body wall muscles and forward-driving motor neurons, and if so, they would involve unpredicted

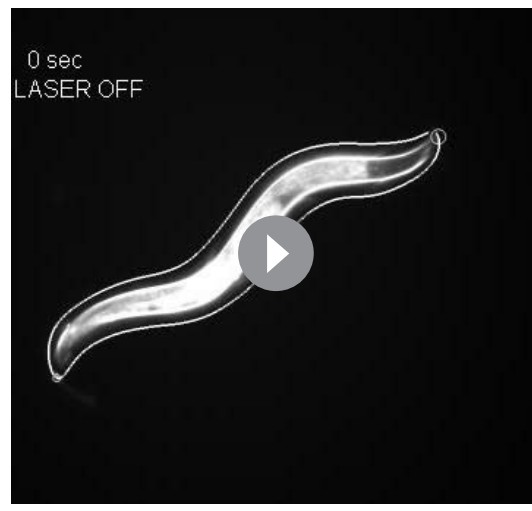

**Video 6.** Representative video of a young adult ZM9315 (RID-specific Chrimson) animal on a thin NGM plate without food, upon RID optogenetic stimulation while the animal was executing forward movement. Head is labeled by the circle on the top right at the beginning of the video.

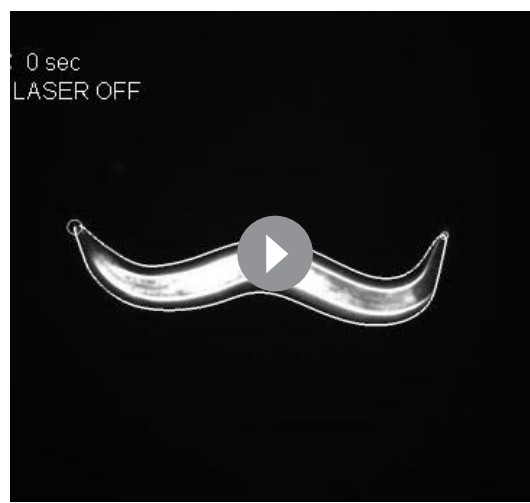

**Video 7.** Representative video of a young adult ZM9315 (RID-specific Chrimson) animal on a thin NGM plate without food, upon RID optogenetic stimulation during reversals. Head is labelled by the circle at the left side at the beginning of the video.

receptors. Screening for GPCR receptors that exhibit physical interactions with FLP-14 offers a plausible entry point.

## Causes for the phenotypic difference between removing RID and FLP-14

In our behavioral assays, RID-ablated animals bear close resemblance to the characteristics and severity of the motor defects exhibited by our *unc-39* mutants. *flp-14* mutants, on the other hand, exhibited phenotypes of the same characteristics but with visibly reduced severity. Such a difference indicates that either RID functions through both FLP-14 and additional effectors to promote forward runs (a possibility that is also consistent with the reduced, instead of abolished motor response upon RID optogenetic stimulation in *flp-14* mutants), and/or, FLP-14 released from additional neurons exert an opposing effect on the forward runs.

We have excluded the possibility of a major involvement of INS-17, the other abundantly expressed neuropeptide by RID, because *ins-17* mutants did not exhibit obvious motor defects. We also tested a possibility that defective GABA uptake may also contribute to the influence of RID on forward runs. RID expresses *snf-11*, a plasma membrane GABA transporter (*Mullen et al., 2006*), suggesting that RID could take up GABA, and either modulate motor output by removing GABA from the surroundings, or by re-releasing the acquired GABA. However, we found that *snf-11* mutants did not show changes in forward and reversal propensity and *snf-11; flp-14* double mutants exhibited the similar motor characteristics as *flp-14* single mutants. PDF-1, a neuropeptide with a known effect on forward runs, was suggested to be expressed by RID (*Janssen et al., 2010*). However, our examination of a PDF-1 reporter strain from a recent study (*Sammut et al., 2015*) showed that the *Ppdf-1*-positive neuron was instead ALA; a notion that also explained why in our datasets, *pdf-1* was an equally enriched transcript from both wild-type and *unc-39* animals (*Supplementary file 1*). NLP-34 could be another RID effector despite its fairly low abundance and less significant enrichment (*Table 1*), but a lack of *nlp-34* mutants prevented us from examining its involvement. In summary, we have not been able to identify other effectors from available reagents.

These negative results would be consistent with another possibility: FLP-14 released from other neurons opposes the effects of FLP-14 released from RID. The release of the same neuropeptide from different neurons can elicit distinct actions on network activity due to different co-transmitters in different neurons, or spatial compartmentalization of the respective signals. Indeed, our *Pflp-14* reporter exhibited expression in several additional neurons, some of which have been implicated in promoting pauses (ALA) (*Fry et al., 2014*) or inhibiting reversals (AIY) (*Gray et al., 2005*). Moreover, we have found that simultaneous optogenetic stimulation of all *Pflp-14*-positive neurons induced immediate reversals, instead of promoting forward runs.

We propose that while RID secretes FLP-14 as a key effector in modulating motor circuit output, RID may also function through additional co-effectors, and FLP-14 may play opposing roles via different neurons. Because most *C. elegans* neurons likely co-express classical neurotransmitters and neuropeptides, and the latter may interact with multiple receptors, an incomplete phenotypic recapitulation upon the perturbation of neurons, neuropeptides that they produce, and receptors that they act upon, may not be uncommon.

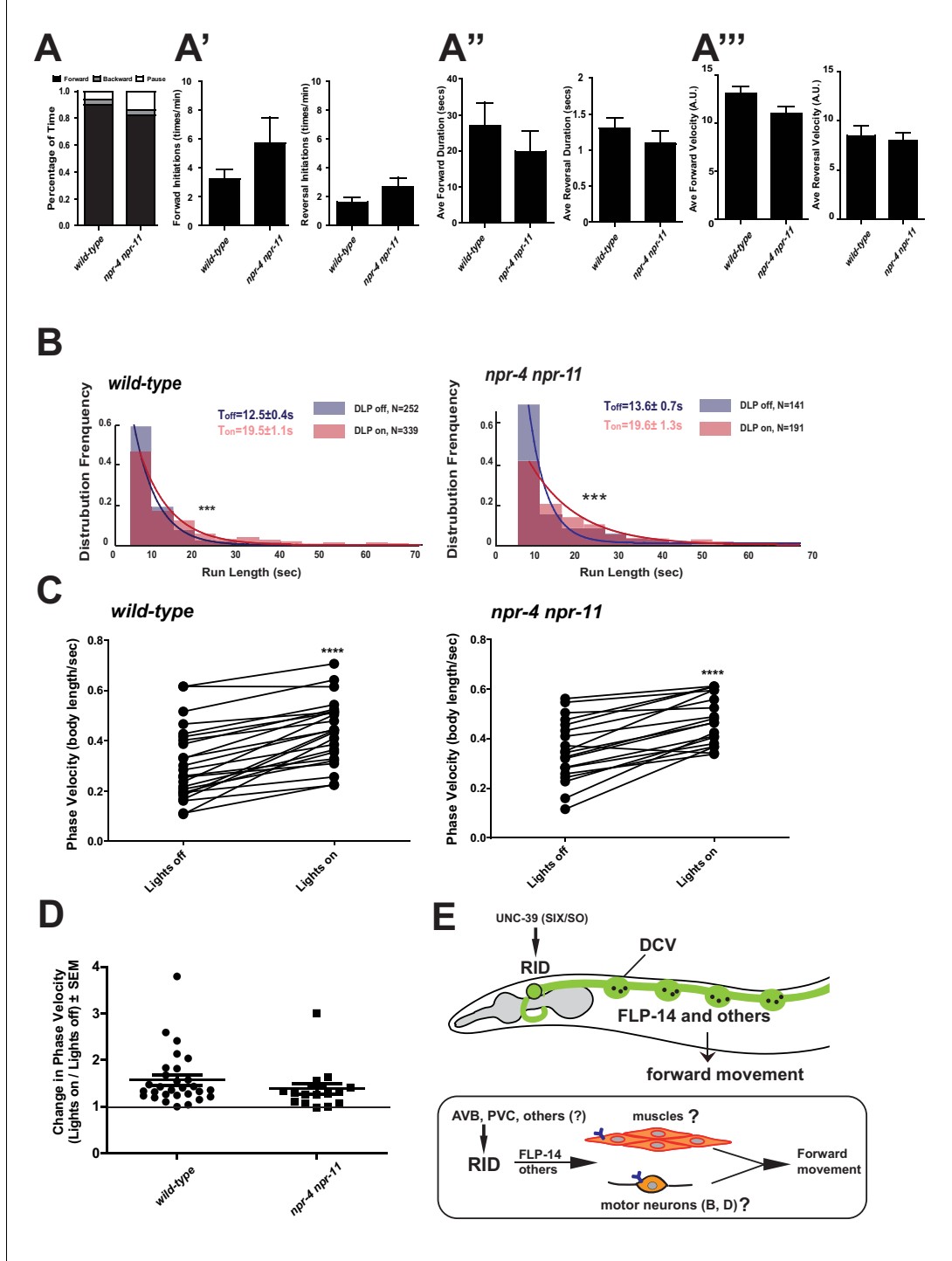

**Figure 8.** FLP-14 may not function through predicted GPCR receptors. (A–A''') Spontaneous motor output, the propensity of directional movement, and the interruption of forward movement between wild-type (N2) and *npr-4 npr-11* animals. Unlike the case for *flp-14* and *unc-39* mutants, or RID-ablated animals, *npr-4 npr-11* double mutants did not exhibit significant motor behavioral changes. N = 10 animals/per genotype. Error bars are ± SEM. (B) The distribution of the mean run length for all light ON (RID stimulation) and light OFF (no RID stimulation) periods. (C, D) A comparison of the motor behavior response before and after RID optogenetic stimulation in *npr-4 npr-11* mutants. The change of speed (phasic velocity) before and after RID stimulation in *npr-4 npr-11* animals was quantified in D. Wild-type and *npr-4 npr-11* double mutants showed a similar increase of run length and velocity in response to RID stimulation. (E) A schematic summary of our findings on RID's role in sustaining forward locomotion (upper panel), and a speculative model of its currently unknown circuit mechanism (lower panel).

*Figure 8 continued on next page*

*Figure 8 continued*

The following figure supplement is available for figure 8:

**Figure supplement 1.** The loss of PVC or AVB alone does not abolish RID activity rise during forward movement.

## A role for the Six family of transcription factors in mitosis and endocrine differentiation

The UNC-39/Six/SO family transcription factors regulate development in various animals and tissues, and human mutations in Six genes are associated with multiple disorders including cancer (*Ford et al., 1998*), holoprosencephaly (*Roessler and Muenke, 1998*), branchio-oto-renal syndrome (*Hoskins et al., 2008*), and Human Myotonic Dystrophy Type 1 (*Harris et al., 2000*). Our study pinpoints a critical, lineage-specific role of a Six family transcription factor in mitosis, a cellular function that may unify and underlie vastly diverse phenotypic presentations. In *C. elegans*, while multiple cells are affected by *unc-39* mutations, the RID lineage is particularly sensitive to the perturbation of UNC-39 activity. A key feature of the RID lineage defect was the random timing of the second round of mitosis in precursor cells. Such a phenotype implies a crucial role for UNC-39 in the onset of and exit from the cell cycle that is to be followed by RID terminal differentiation. Mechanisms that couple cell cycle and terminal differentiation have remained elusive (*Hardwick and Philpott, 2014*). We propose that UNC-39 and its RID lineage-specific partners may be responsible for such a coupling. The absence of corpora cardiac, the insect pituitary in *Drosophila SO* mutants further entices the speculation of functional conservation for the Six transcription factors in endocrine development (*De Velasco et al., 2004*).

The molecular profiles of neurons of diverse lineages facilitate the exploration of the cellular properties that endow their synaptic connectivity and functional specificity, as well as their evolutionary origins. Animal models such as *C. elegans* and *Drosophila* have a rich repertoire of genetic tools including mutants with specific and known lineage changes (*Hobert, 2011*). Methods to collect sufficient quantities of fully differentiated neurons from intact *C. elegans* for mRNA sequencing have been developed (*Kaletsky et al., 2016*; *Spencer et al., 2014*; *Wang et al., 2015b*). By comparing the transcriptome profiles obtained from specific cells from wild-type animals and *unc-39* mutants, we have established a robust pipeline that led to the identification of both reported and previously unknown RID-enriched transcripts (*Supplementary files 1* and *2*). Some of these transcripts represent candidate transcripts either activated or repressed by UNC-39. A well-controlled subtractive transcriptome dataset should provide not only cell-type markers, but also targets that are regulated by lineage-specifying transcription factors. The functional validation of neuropeptides obtained from this dataset suggests its benefit and utility for other studies.

## Materials and methods

### Strains and constructs

*C. elegans* were cultured on standard Nematode Growth Medium (NGM) plates seeded with OP50, and maintained at 22°C. *hp701* was isolated by EMS mutagenesis of a strain carrying an integrated *Pceh-10-GFP* array (*hpIs202*), identified through a combination of SNP mapping and whole genome sequencing (*Davis and Hammarlund, 2006*; *Doitsidou et al., 2010*), followed by rescuing by a *unc-39*-containing fosmid and a *unc-39* minimal genomic clone. *hp701* and other mutants obtained from the *Caenorhabditis* Genetics Center (CGC) were backcrossed at least 4x against N2 prior to usage. See Supplemental Methods for a complete list of mutants, transgenic lines, and constructs.

### Electron microscopy

L1 (N2) or young adult animals (N2 and *hp701*) were packed into 3 mm diameter aluminum carriers and frozen using a Leica EM HP100 high-pressure freezer. Subsequent fixation and freeze substitution was performed as described (*Weimer, 2006*). Samples were infiltrated and embedded in the Spurr resin. 70 nm transverse serial sections were prepared and imaged using the FEI Techai 20 TEM at 200 kV (x29,000). For L1, the entire length of RID axon along the dorsal cord was reconstructed.

For adults, 20–30 µm of the dorsal nerve cord was reconstructed. See Supplemental Information on EM reconstruction and analyses.

## Embryonic lineage analysis

Strains (*hpIs202* and *unc-39; hpIs202*) were cultured and recorded at 25°C. Embryos were mounted at 1–4 cell stages, and recorded every 35 s with 25 z-levels. After ~350 z-scans (200 min) with DIC optics, the GFP fluorescence optics was automatically activated and exposed for every 20th z-scan till 320 min of development, and then every 30th z-scan till the three-fold stage. Recording was performed at 630x on a 4D microscope controlled by *TimeToLive* (*Caenotec*-Prof. Ralf Schnabel r.schnabel@tu-bs.de). RID and neighboring lineage cells were traced using SIMI°Biocell (SIMI Reality Motion Systems GmbH, http://www.simi.com/de/home.html) until the late 1.5-fold stage (340 and 360 min) as described (*Schnabel et al., 1997*).

## Preparation of dissociated larval cells for cell sorting

Synchronized *hpIs202 (Pceh-10-GFP)* and *unc-39(hp701); hpIs202* larval stage 1 (L1) animals were grown on 150 mm 8P plates seeded with NA22 for approximately ~12 hr until they reached larval stage 2 (L2). Cells were extracted from L2s. See Supplemental Information for detailed methods.

## Laser ablation

Synchronized *hpIs202 (Pceh-10-GFP)* L1 animals were immobilized and RID ablated using the Micro-Point laser as described (*Fang-Yen et al., 2012*). Mock-ablated animals were L1s of the same genotype processed similarly, except that they were not exposed to the laser. L4 RID- and mock-ablated animals were confirmed for the status of RID prior to locomotion analyses.

## Locomotion assay

Automated tracking and locomotion analyses were performed as described (*Gao et al., 2015*). Detailed description of experimental conditions is provided in the Supplemental Information.

## Calcium imaging

The third stage (L3) *hpIs587 (Pflp-14-GCaMP6::cherry)* animals were placed on a drop of M9 buffer on top of a 2.5% agar pad, covered with a cover slip. Four ~1 mm × 1 mm 0.001 inch-thick polycarbonate pieces (Catalog # 9513K12, McMaster-Carr) between the agar pad and coverslip served as spacers to allow movement. Each recording lasted for 3 min. Images were captured using a 20x objective on a Zeiss Axioskop 2 Plus equipped with an ASI MS-40000 motorized stage, a dual-view beam splitter (Photometrics) and a CCD camera (Hamamatsu Orca-R2). The 4x-binned images were obtained at 10 frames per second. See Supplemental Information for detailed methods.

## Optogenetic stimulation

We restricted the expression of a codon-optimized version of Chrimson to the RID neuron, and stimulated wild-type, *flp-14(gk1055)*, *unc-39(hp701)* and *npr-4(tm1782)npr-11(ok594)* animals carrying the same transgene using a Diode red laser while animal locomotion was tracked under dark-field imaging by infrared light. The stimulation and tracking were performed using the Colbert (COntrol Locomotion and BEhavior in Real Time) system (*Leifer et al., 2011*). The locomotion was recorded for each animals, and their trajectories were then extracted and analyzed as described analyzed as previously described (*Luo et al., 2014*). See Supplemental Information for detailed methods.

## Electron microscopy analysis

Images were stitched and aligned using TrakEM2 (*Cardona et al., 2012*). In the L1 animal, all neurons were identified based on cell body position (*Sulston et al., 1983*), synapse pattern and neurite trajectory (*White et al., 1986*). In the adult EM reconstruction, all neurons were identified by characteristic synapse patterns and trajectories (*White et al., 1986*). Volumetric reconstruction of neurons was performed using TrakEM2 and skeleton tracing using CATMAID (*Saalfeld et al., 2009*) followed by rendering in Blender (http://www.blender.org).

## Fluorescence microscopy and confocal imaging

For imaging dense core vesicle (*Pceh-10-IDA-1::cherry*) and neuropeptide fluorescent markers (*Pceh-10-INS-22::GFP*), expression patterns of UNC-39 (*Punc-39-GFP, Punc-39-UNC-39::GFP*) and RID phenotypes in *unc-39* mutants using RID fluorescent markers (*Pceh-10::GFP*) and RID cell fate markers (*Pkal-1-GFP, Pser-2-GFP, Pmod-1-GFP*), images were captured using a 63x objective on a Zeiss Axioplan two connected to a Hamamatsu ORCA-ER digital camera and processed using Improvision Open Lab software. Images were processed using minimal deconvolution levels to remove background fluorescence. Confocal images of transgenic strains carrying either *Pflp-14-GFP*, *Pins17-GFP*, *unc-3fosmid::SL2::GFP*, and *unc-39fosmid::GFP* were acquired on a Nikon Eclipse 90i confocal microscope. Confocal image processing was conducted using Adobe Photoshop. The primer pair to generate a 4.3 kb *Pflp-14* fragment are tactgtcgaccgacaaacacccaaatatcc (forward, SalI) and aactggatcctccttcggattgtgtggag (reverse, BamHI).

## Larval cell extraction and cell sorting

Briefly, synchronized animals were pelleted and thoroughly washed with M9 Buffer to remove bacterial contamination. Prior to extraction, an aliquot of sample was flash-frozen in liquid nitrogen to be used as the All Cells reference sample for subsequent transcriptome analysis. To the remaining sample, freshly thawed SDS-DTT solution (0.25% SDS, 200 mM DTT, 20 mM HEPES, 3% sucrose pH = 7.5–8.0), which softens the cuticle, was added for no longer than 4 min (or until majority of animals started twitching, but did not become completely rod-like or rigid). Post-incubation with SDS-DTT, the sample was neutralized and washed 5x times with 1x Egg Buffer. To break apart animals, Pronase solution (15 mg/mL) was added, and the sample was pipetted ~40x for 30 min. After several washes, cells were resuspended in 1x Egg buffer. Propidium iodide (1 μg/mL) was added prior to cell sorting to identify damaged cells.

GFP+ cells from L2 worms were sorted onto Trizol-LS (Invitrogen) using a BD FACSAria with a 70 micron nozzle (BD Biosciences) operated at the University of Toronto Flow Cytometry Facility. Profiles of GFP+ strains were compared to an N2 standard to identify and exclude autofluorescent cells. For each independent replicate, 20,000–50,000 events, most likely representing cells, were FACS-isolated from each strain. Three and four biological replicates with corresponding All Cells reference samples were collected for the *hpIs202* control and *the unc-39 (hp701); hpIs202* strains, respectively.

## RNA-sequencing preparation and analysis

### RNA extraction

For the all cells reference samples, flash-frozen pellets were pulverized using a mortar and pestle and dissolved in Trizol-LS (Invitrogen). RNA was extracted from these pulverized worm pellets (All and from sorted cells collected directly in Trizol-LS (GFP+ cells sample). DNA contamination was removed using the Zymo DNA-free RNA Kit (Zymo Research, Irving, CA) according to manufacturer's instructions.

### Library preparation and RNA-sequencing

RNA sample concentration and quality were determined using an Agilent Bioanalyzer at The Centre for Applied Genomics (SickKids Hospital, Toronto, ON, Canada). RNA Integrity Numbers (RIN) scores of seven and above were used for subsequent RNA-sequencing analysis. RNA concentration was also verified using the Quibit RNA HS Kit (Thermo Fisher Scientific).

5 ng of RNA was used as starting material for cDNA library preparation. cDNA was synthesized from RNA using the SMARTer Ultra Low Input RNA Kit for Sequencing (Clontech) and cDNA libraries prepared using the Low Input DNA Library Prep Kit (Clontech) according to manufacturer's instructions. RNA-sequencing was performed on an Illumina Hiseq 2000 according to standard protocols, generating 100 base paired-end reads.

### Bioinformatics analysis

Sequencing reads were mapped to the *C. elegans genome* (WS235) using RNA STAR under default settings (*Dobin et al., 2013*). Using these conditions on RNA STAR, 75–80% of reads aligned to a unique transcript. Gene expression quantification and differential expression were analyzed using

HTSeq (*Anders et al., 2015*) and DESeq (*Anders and Huber, 2010*), respectively, under default settings.

Using DESeq, we identified differentially expressed transcripts between GFP+ cells and All Cells reference samples in the *hpIs202* and *unc-39(hp701);hpIs202* datasets. For *hpIs202* and *unc-39 (hp701);hpIs202* datasets, transcripts were considered significantly enriched in GFP+ cells over sample matched All Cells by applying the following criteria or filters: (1) Differentially expressed transcripts with False Discovery Rate (FDR) adjusted p values<0.05, and, (2) Transcripts where the ratio of log2 transformed Mean Counts from GFP+ samples to log2 transformed Mean Counts from All Cells samples was greater than 1 (Mean GFP+ Counts / Mean All Cells Counts >1). Final datasets resulted from analyses of >3 experimental replica from All cells and GFP+ samples for wild-type and *unc-39* mutant strains.

## Validation of the datasets

Our final datasets were validated using >3 replicates of the wild-type samples. We assessed its quality by two criteria: first, the identification of transcripts known to be expressed in RID, ALA, AIY, or CAN (*Supplementary file 1*; Positive Controls). We detected significant enrichment for transcripts reported to be expressed by either all GFP+ cells (*Svendsen and McGhee, 1995*), or a subset of GFP+ cells, e.g. in AIY (*ttx-3*, *hen-1*, *glc-3*, *pdfr-1*) (*Flavell et al., 2013*; *Ishihara et al., 2002*; *Wenick and Hobert, 2004*), and in RID (*lim-4*, *snf-11*, *zig-5*, *ser-2* and *kal-1*) (*Bülow et al., 2002*; *Mullen et al., 2006*; *Tsalik and Hobert, 2003*); Second, the absence of transcripts from non-neuronal tissues, such as muscles (*myo-2*, *myo-3*) or neuronal subtypes not included in GFP+ Cells - the glutamatergic (*eat-4*) and GABAergic (*unc-25*) neurons (*Supplementary file 1*; Negative Controls). Lastly, this study validated the presence of FLP-14 and INS-17 in RID (*Table 1*) by reporter analyses (*Figure 3*) and the functional relevance of FLP-14 (*Figure 6*).

## Locomotion behavior analysis

### Behavior acquisition and tracking

When transferred to a new, thinly seeded plate, *C. elegans* typically spend most of the time moving forward, with brief interruptions of backward movement. As previously described with some modifications (*Gao et al., 2015*), 35 mm Nematode Growth Media (NGM) plates with limited food (lightly seeded OP50 bacteria) were used for automated tracking and behavioral analyses. Using this method, we quantified the percentage of time animals spent moving forward, backward, and pausing. We also quantified initiation frequency, duration, and velocity of larval stage 4 (L4) animals. For RID-ablated animals, controls were mock-ablated animals carrying the same GFP reporter. For *unc-39*, *flp-14*, *ins-17*, and *npr-4 npr-11* genetic mutants, controls were N2 animals.

Prior to recording, animals were placed in the center of a 50 mm lightly seeded NGM plate and allowed to habituate for 5 min prior to recording. Behavior was recorded for 3 min under a 40x objective using a 20Zeiss Axioskop 2 Plus equipped with an ASI MS-40000 motorized stage and a CCD camera (Hamamatsu Orca-R2).

Tracking and analysis were performed using Micromanager and ImageJ software plugins developed in-house (courtesy of Dr. Taizo Kawano). Image sequences were sampled at 100-msec exposure (10 frames per second). The directionality of movement (forward vs. backward) was determined by first identifying the anterior-posterior axis or the 'head' and 'tail' points, which were manually defined at the first two frames and verified throughout the recording. To calculate directionality of movement, the displacement of the midline point in relation to the head and tail for each worm was determined based on its position in the field-of-view and the stage coordinates. Image sequences where animals touched the edge of the recording field or crossed over on themselves were not processed.

### Quantification and data analyses

Analyses of the output data were carried out using an R-based code developed in-house (courtesy of Dr. Michelle Po). The following parameters were quantified by the program: (1) Initiation (defined as the frequency of directional change for each animal); (2) Duration (defined as the time spent moving in the same direction for >3 frames or 300 msec, calculated for each bout of forward or reversal initiation); (3) Velocity (defined by the speed, displacement of animal divided by the # of frames, and

directionality of the animal). Frequency of initiations, durations, and velocities were calculated for forward and reversal locomotion separately.

## Calcium imaging and data analysis

Regions of interest (ROIs) containing the RID neuron was defined using a MATLAB script developed in-house. GCaMP and RFP fluorescence intensities from RID were then measured. To analyze overall RID activity, the ratio of GCaMP to RFP was calculated in order to control for possible motion artifacts detected. The velocity of each time point was measured using an Image J plug-in developed in-house (*Gao et al., 2015*; *Kawano et al., 2011*). The rate for Acceleration during each transition from backward to forward locomotion was calculated by subtracting the lowest velocity point during backward locomotion from the highest velocity point during forward locomotion, and normalizing it to the number of frames in-between these two points (Velocity$_{peak}$ – Velocity$_{trough}$ / # of Frames). The rate of Deceleration during transitions from forward to backward locomotion was calculated similarly (Velocity$_{trough}$ – Velocity$_{peak}$ / # of Frames). Rise and decay in calcium transients during transitions from backward to forward locomotion and vice versa, respectively, were calculated using the linear slope. Cross-correlation analyses were performed between rate of acceleration and calcium rise, as well as deceleration and calcium decay.

## Optogenetic stimulation and data analyses

To restrict expression of the Chrimson protein to RID, we drove the expression of a Chrimson::GFP: ZF construct by *Pceh-10*, in a subset of neurons (RID, ALA, AIY, and CAN), and the expression of ZIF-1::SL2::GFP by *Pgpa-14*, *Pttx-3*, and *Parr-1*, which overlap with *Pceh-10* for ALA, AIY, and CAN, respectively. ZIF-1 targets chrimson::GFP::ZF for degradation. GFP and RFP were used to confirm the specificity of expression pattern of these constructions in targeted neurons.

We first generated an integrated transgene *hpIs626* from an Ex array made from co-injected *Pceh-10-chrimson::GFP::ZF*, *Pgpa-14-ZIF::SL2::RFP*, *Pttx-3-ZIF::SL2::RFP* and the lin-15 co-injection marker. This transgene exhibits restricted chrimson::GFP expression in RID and CAN. *Parr-1-ZIF-1:: SL2-RPF* was then injected in *hpIs626*;; ~10% of the double transgenic animals that carries the Ex array (*hpIs626;hpEx3808*) exhibited RID-specific chrimson::GFP expression. These array were then crossed into *unc-39*, *flp-14*, and *npr-4 npr-11* backgrounds.

For optogenetics experiments, we first picked late larvae to young adult transgenic animals by selecting for those with RID-specific GFP expression onto the OP-50 retinal plates under the dissecting fluorescent microscope. After 24 hr incubation, each animal was individually recorded using the COLBERT stimulation system (see below). After recording, each animal was mounted on slides to be examined for GFP signals; data collected from animals that exhibited residual GFP signals in CAN were discarded.

To make OP50-retinal plates, we seeded each 60 mm NGM plate with a mixture of 250 μL OP50 in LB with 1 μL of 100 mM retinal in ethanol. Animals were then individually washed in the NGM buffer before allowed to navigate on the surface of a 100 mm NGM plate without food. We used the Colbert system to stimulate Chrimson with a Diode red laser (MRL-III-635, 635 nm wavelength, 200 mW maximum power, CNI Laser) and carry out dark-field imaging using infrared light and a dark field condenser (Nikon) using a 10X Plano Apo objective (NA = 0.45). A motorized stage and custom real-time computer vision software kept the animal in the center of the field of view. The locomotion of each animal was recorded for 30 min or until it reached the edge of the plate. The laser was set at a 3 min on/3 min off cycle during the recording. Data was collected at 30 f/s and analyzed using a customized particle-tracking and shape analysis algorithms. Each trajectory was segmented into periods of forward movement (runs) separated by sharp orientations (turns). Turns were automatically flagged when the heading change of the center of the mass trajectory was >60° over 1 s. To quantify the change of phase velocities before and after the laser was switched on (*Figure 6A,B*), we quantified velocity when the laser was on during a run with 10 s of forward movement before and after the switch for wild-type, *flp-14* and *npr-4 npr-11* animals, and 4 s before and 4 s after the switch for *unc-39* animals. To quantify the forward run length, all runs during the laser on phase and laser off phase were included.

## Statistical analysis

For locomotion (# of initiations, durations, velocities), statistical significance was determined using Mann-Whitney or Kruskal-Wallis tests for comparing two and more than two variables, respectively, and subjected to post-hoc analysis. For calcium imaging (correlations between acceleration/deceleration and rise/decay of calcium transients), Pearson tests were used to calculate correlation coefficients and statistical significance. For optogenetic stimulation and locomotion analyses, Wilcoxon matched-pairs signed rank test was used to compare the velocity difference during periods of Lights On and Off for the same strain. The Kruskal-Wallis Test with post-hoc test (Dunn's Multiple Comparison) was used to assess the significance of difference between groups. $p < 0.05$ were considered to be statically significant. All statistics were performed using Prism software (GraphPad).

## Acknowledgements

We thank D Witvliet, Y Wang and Q Jin for technical assistance and EM analysis; *CGC*, H Hutter and O Hobert for strains, D Moerman and S Flibotte for whole-genome sequencing analyses. This work was supported by CIHR and NSERC (to MZ), NIH (to JAC, ADTS and DMM), and NSFC (to LL).

## Additional information

### Funding

| Funder | Grant reference number | Author |
|---|---|---|
| National Natural Science Foundation of China | 11304153 | Linjiao Luo |
| National Institutes of Health | ns082525-01A1 | Aravinthan DT Samuel |
| Human Frontier Science Program | RGP0051/2014 | Aravinthan DT Samuel<br>Mei Zhen |
| National Institutes of Health | R01NS079611, R01NS081259 | David M Miller |
| National Institutes of Health | DP5OD009152 | John A Calarco |
| Canadian Institutes of Health Research | MOP93619, MOP123250 | Mei Zhen |
| Natural Sciences and Engineering Research Council of Canada | 262112-12 | Mei Zhen |

The funders had no role in study design, data collection and interpretation, or the decision to submit the work for publication.

### Author contributions

MAL, BM, YLi, Acquisition of data, Analysis and interpretation of data, Drafting or revising the article; JC, VL, DF, LL, YLu, C-YH, DH, NJ, Acquisition of data, Analysis and interpretation of data; JM, RM, DMM, RS, Analysis and interpretation of data, Drafting or revising the article; AW, Acquisition of data, Analysis and interpretation of data, Drafting or revising the article; WH, Acquisition of data, Drafting or revising the article; YQ, Acquisition of data, Contributed unpublished essential data or reagents; ADTS, Analysis and interpretation of data, Drafting or revising the article; JAC, MZ, Conception and design, Analysis and interpretation of data, Drafting or revising the article

## Additional files

### Supplementary files

• Supplementary file 1. Table of control transcripts.

• Supplementary file 2. Table of enriched transcripts in wild-type and unc-39 cells.

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

## Appendix 1

### Additional tables

**Appendix 1—table 1.** A list of genetic mutant strains generated and/or used.

| Gene | Allele | Strain |
|------|--------|--------|
| *unc-39 V* | *hp701* | ZM6539 |
| *unc-39 V* | *e257* | CB257 |
| *ced-3 IV* | *n717* | ZM6097 |
| *ced-4 III* | *n1162* | ZM6098 |
| *unc-3 X* | *xd86* | ZM9343 |
| *flp-14 III* | *gk1055* | ZM8969 |
| *ins-17 III* | *tm790* | ZM2860 |
| *Ins-17 flp-14 III* | *tm790 gk1055* | ZM9030 |
| *snf-11 V* | *ok156* | RM2710 |
| *flp-14III; snf-11V* | *gk1055; ok156* | ZM9215 |
| *npr-4 X* | *tm1782* | ZM9455 |
| *npr-11 X* | *ok594* | ZM9454 |
| *npr-4 npr-11 X* | *tm1782 ok594* | ZM9455 |

**Appendix 1—table 2.** A list of constructs and transgenic strains generated and used (LIN-15 was used as an injection marker if not specified).

| Experiment | Plasmid | Description (injection marker) | Background | Transgene | Strain |
|------------|---------|-------------------------------|------------|-----------|--------|
| *RID Reporters* | pJH2103 | *Pceh-10::GFP* | *lin-15(n765)* | *hpIs202* | ZM5488 |
| | | | | *hpIs201* | ZM5489 |
| | pJH1647 | *Pceh-10::Cherry* | *lin-15(n765)* | *hpIs292* | ZM6905 |
| | pJH1647 | *Pceh-10::Cherry (pRF4)* | *juIs1* | *hpIs162* | ZM8000 |
| | pJH2160 pJH2247 | *Pceh-10::IDA-1::Cherry Pceh-10::ins-22::GFP* | *lin-15(n765)* | *hpEx3669* | ZM8823 |
| | pJH2247 pJH2160 | *Pceh-10::ins-22::GFP Pceh-10::IDA-1::Cherry* | *lin-15(n765)* | *hpEx3669* | ZM8823 |
| | Hobert lab | *Pkal-1::GFP* | *unknown* | *otIs33* | OH904 |
| | Hobert lab | *Pser-2::GFP* | *unknown* | *otIs107* | OH2246 |
| | pJH2715 | *Pmod-1::mito::GFP* | *lin-15(n765)* | *hpIs274* | ZM6658 |
| | Huang lab | *unc-3 fosmid::SL2::GFP (sur-5::RFP)* | *unc-3(xd86); hpIs162* | *xdEx1091* | XD2319 |

*Appendix 1—table 2 continued on next page*

Appendix 1—table 2 continued

| Experiment | Plasmid | Description (injection marker) | Background | Transgene | Strain |
|---|---|---|---|---|---|
| unc-39 Related | pJH2765 | Plim-4::lim-4::GFP | lin-15(n765) | hpEx3035 | ZM7100 |
| | pJH2839 | Punc-39::GFP | lin-15(n765) | hpIs328 | ZM7150 |
| | pJH2798 | Genomic unc-39 minimal rescuing clone | unc-39 (hp701); | not maintained | not maintained |
| | pJH3138 | unc-39fosmid::GFP (Pmyo-3::RFP) | unc-39(e257) | hpEx3186 | ZM7572 |
| | pJH2811 | Punc-39::UNC-39::GFP (RF4) | unc-39 (hp701); hpIs292 | hpEx3034 | ZM7482 |
| | pJH2839 | Punc-39::GFP | lin-15(n765) | hpIs328 | ZM7150 |
| | pJH3366 | Punc-39::unc-3 cDNA (RF4) | unc-39 (hp701); hpIs202 | hpEx3498 | ZM8312 |
| | pJH3084 pJH3366 | Punc-39::LIM-4 cDNA::GFP Punc-39::UNC-3 cDNA | unc-39 (hp701); hpIs292 | not maintained | not maintained |
| flp-14 Related | pJH3884 | flp-14 genomic miniMos (NeoR) | N2 (wt) | hpSi38 | ZM9518 |
| | | | flp-14(gk1055) | hpSi38 | ZM9474 |
| | | | flp-14 (gk1055); unc-39(hp701) | hpSi38 | ZM9468 |
| | | | hpIs201 | hpSi38 | ZM9473 |
| | | | flp-14 (gk1055); hpIs201 | hpSi38 | ZM9519 |
| | pJH2103 | Pceh-10::GFP | flp-14(gk1055) | hpIs201 | ZM9502 |
| flp-14 and ins-17 Reporters | pJH3608 | Pflp-14::GFP | lin-15(n765) | hpEx3695 | ZM8935 |
| | Hutter lab | Pins-17::GFP (unc-119) | unc-119(ed3) | wwEx73 | HT1734 |
| Calcium Imaging | pJH3644 | Pflp-14::GCaMP6::Cherry | lin-15(n765) | hpIs587 | ZM9078 |
| | | | hpIs321 | hpIs587 | ZM9312 |
| | | | juIs440 | hpIs587 | ZM9404 |
| Optogenetic Stimulation | pJH3790 pJH3796 pJH3774 | Pceh10::Chrimson::GFP::ZF Pttx-3::ZIF-1::SL2::RFP Pgpa-14::ZIF-1::SL2::RFP | lin-15(n765) | hpIs626 | ZM9331 (RID/CAN) |
| | | | hpIs626 | hpEx3808 | ZM9351 (RID only) |
| | pJH3835 | Parr-1-ZIF-1::SL2::RFP | unc-39 (hp701); hpIs626 | hpEx3808 | ZM9472 (RID only) |
| | | | flp-14 (gk1055); hpIs626 | hpEx3808 | ZM9476 (RID only) |
| | | | npr-4(tm1782) npr-11(ok594); hpIs626 | hpEx3808 | ZM9529 (RID only) |

