## [Decision Letter]

Thank you for submitting your article "Neuroendocrine Modulation Sustains the *C. elegans* Forward Motor State" for consideration by *eLife*. Your article has been reviewed by three peer reviewers, and the evaluation has been overseen by Eve Marder as the Senior Editor and Reviewing Editor. The following individual involved in review of your submission has agreed to reveal his identity: Akinao Nose (Reviewer #2).

The reviewers have discussed the reviews with one another and the Reviewing Editor has drafted this decision to help you prepare a revised submission.

In this manuscript the authors go to significant length to argue that a newly identified neuroendocrine neuron is important in behavior. All of the reviewers found the manuscript interesting, but have some specific issues that need addressing.

Essential revisions:

Most notably, please move Figure 7 to the Results section. Also, please revise the manuscript, using the reviews as a guide, to distinguish clearly what you have done and what is a speculative conclusion. Speculations and a speculative cartoon model are fine for the Discussion as long as the reader is clearly told what are findings and what are speculations.

I am leaving portions of the initial reviews for your information because I don't think they are in disagreement and might be informative to you.

*Reviewer #1:*

This manuscript takes the novel approach of identifying possible neuroendocrine cells by sTEM and determining their functions. The authors identify the RID neuron as a neuroendocrine cell because of its large number of dense core vesicles. In a screen for mutants that specifically disrupt RID differentiation, they identified the Six/SO homeodomain transcription factor UNC-39. Loss of unc-39 results in loss of several RID markers and its distinctive axonal morphology. Although the unc-39 results are solid, they were not well integrated into the overall theme of the paper (neuropeptide modulation of forward movement).

During forward movement, calcium levels in RID increased, whereas forward to reverse transitions caused a decrease in calcium levels, suggesting that RID is involved in forward movement. The calcium imaging was supported by behavioral work that showed that unc-39 mutants and RID-ablated animals had shorter forward runs and more turns than wild type, although unc-39 mutants had a more severe phenotype than the RID-ablated animals (Figure 1), suggesting that unc-39 regulates other neurons that sustain forward movement. The authors followed up by showing that optogenetic stimulation of RID caused a sustained increase in forward velocity and runs.

The authors performed a subtractive transcriptome analysis to identify three candidate neuropeptide genes that are enriched in RID: flp-14, ins-17, and nlp-34. flp-14 mutants show a decreased average forward run duration, an increased number forward initiations, and an increased number of reversal initiations compared to wild type. However, as the authors note, the defects in flp-14 are not as severe as in unc-39 mutants and RID-ablated animals, suggesting that other peptides may act in concert with FLP-14 to modulate forward movement. In addition, the behavioral assays were highly variable (i.e., the SEMs are very large, suggesting a lot of variability in the behaviors). For instance, the wild-type values for average forward duration were significantly different in Figure 5. This variability makes looking at defects in flp-14 mutants and its rescue difficult; for instance, 1) the average forward duration for flp-14 mutants looked significantly different from wild type in Figure 5, but not really in Figure 5; 2) the flp-14 mutants were not rescued with a copy of the flp-14 transgene for average forward duration to levels seen in wild type or even the flp-14 transgenics alone (Figure 5); and 3) the flp-14 mutants did not look significantly different from the flp-14 transgenic line (Figure 5). The only consistent defect seen with the flp-14 mutants was the decrease in the number of forward initiations. The authors should re-word some of their statements.

The authors indicate that ins-17 mutants acted similarly as wild type and the double mutants acted similar to the flp-14 mutants alone. However, flp-14 ins-17 double mutants showed more severe phenotypes than flp-14 mutants alone (Figure 5—figure supplement 1), contrary to what the authors indicate, suggesting that INS-17 peptides might also be contributing to forward movement. Their statement should be re-worded.

Analysis of the possible FLP-14 receptors was not very informative. Double knockout of the possible receptors, NPR-4 and NPR-11, showed phenotypes that were not significantly different than wild type, but showed the same general trend as flp-14 mutants. Despite this negative result, Figure 7 should be included in the text rather than in the Discussion.

The paper lacks a model of how RID inputs into the motor circuit to sustain forward motion. RID synapses onto hypodermal cells, DD neurons, and muscle, but could also act humorally. Do the authors hypothesize that the effect of FLP-14 peptides released from RID is on DD neurons, muscle, or other targets? In Ascaris, application of FLP-14 on muscle causes relaxation followed by rhythmic contractions or sustained contractions (depending on which paper one reads). The authors should have a model for the neuromodulation.

*Reviewer #2:*

1) Specificity of the activation experiments with Chrimson. Please explain more in detail how the authors restricted the expression of Chrimson to RID neurons using the ubiquitin ligase system. Since the specificity of Chrimson expression is critical for the interpretation of the results, inclusion of a picture showing the specific expression as a supplementary figure would be desirable. I also wonder how one should interpret the results of the optogenetic activation in unc-39 mutants. Is the reversion of the phenotype due to lack of Chrimson expression in RID or defective function of the neuron?￼￼

2) It is not clear how RID regulates forward locomotion. I would expect more through discussion on putative target cells of RID in relation to known locomotory circuits. An obvious candidate would be DD MNs whose terminals reside in the immediate neighbor of RID varicosities. What happens to forward locomotion when DD MNs are inhibited or activated? If feasible, the authors may also examine if the activity manipulation of RID or application of FLP-14 affects the level of Ca^2+^ or other signaling molecules (e.g., cAMP) in DDs.

*Reviewer #3:*

1) There seems to be a caveat implied by Figure 6 that should be discussed. The speed of unstimulated flp-14 mutants was significantly higher than that of wild-type while the stimulated speeds of the two strains were much more similar. Therefore, a possible (plausible?) interpretation of the difference in fold change is a ceiling effect. The animals, on average, didn't move faster than ~0.55 body-lengths per sec such that the unstimulated mutants didn't have as much capacity for speedup as wild-type. The data in panel C possibly suffers from a similar (not unrelated) issue: T_off for the mutants is higher than wild-type.

If so, the optogenetic assay does not strongly support the conclusions of the manuscript although it doesn't contradict them either. It is perfectly reasonable for a particular assay of a complex phenomenon to fall short of providing conclusive evidence. It is even useful to present such results by way of transparency and completeness. Thus, the data is useful and should be presented in the main text without detracting from the interest of the manuscript or the validity of the conclusions.

If I am mistaken, perhaps the caveat should be briefly mentioned and refuted for the sake of clarity. Otherwise, the caveat should be discussed and the conclusion of the relevant Results section should be revised to state that the optogenetic assays were consistent with the overall theme of the work but in and of themselves inconclusive.

2) Figure 7 depicts a negative result, suggesting that the predicted receptors of FLP-14 do not directly affect forward locomotion (either because the prediction is wrong or because their interaction with multiple ligands is more complex). I think that such data does belong in a Results section. It is of practical importance to *C. elegans* researchers and of conceptual importance to any researcher that felt the need to tuck away a valid result in the name of a 'cleaner', more linear story.

---

## [Author Response]

Essential revisions:

*Most notably, please move Figure 7 to the Results section. Also, please revise the manuscript, using the reviews as a guide, to distinguish clearly what you have done and what is a speculative conclusion. Speculations and a speculative cartoon model are fine for the Discussion as long as the reader is clearly told what are findings and what are speculations.*

*I am leaving portions of the initial reviews for your information because I don't think they are in disagreement and might be informative to you.*

**Response**: Thank you. We gladly moved the figure and related texts to the Results (see Figure 8 in Resubmission; Figure 7 in previous submission). We clarified in figure legends that the upper panel in the model sketch in Figure 8 (Figure 6 in previous submission) summarizes our findings on RID, whereas the lower panel represents the unknown downstream signaling from RID. We do not yet know how RID integrates in the forward circuit, a point that we expanded in Discussion. We feel more comfortable showing our conclusions with minimal speculation, and hope that this is acceptable. We appreciate comments from all reviewers and summarize below our responses and revisions that we made in the re-submitted manuscript.

Reviewer #1:

*This manuscript takes the novel approach of identifying possible neuroendocrine cells by sTEM and determining their functions. The authors identify the RID neuron as a neuroendocrine cell because of its large number of dense core vesicles. In a screen for mutants that specifically disrupt RID differentiation, they identified the Six/SO homeodomain transcription factor UNC-39. Loss of unc-39 results in loss of several RID markers and its distinctive axonal morphology. Although the unc-39 results are solid, they were not well integrated into the overall theme of the paper (neuropeptide modulation of forward movement).*

**Response:** Thank you. The *unc-39* work, including the embryonic lineage analyses that denote its deficiency of RID neurogenesis, its EM analyses, its motor behavioral deficit, and RNA profiling of *unc-39* neurons, could indeed be a stand-alone study. The focus of such a ‘paper’ would have been UNC-39’s requirement for RID’s development. However, the focus of our paper, representing the main interest of my group, is the RID neuron and its role in the motor circuit.

In our study, UNC-39 is an essential tool for three key aspects of RID: 1) RID’s identity as a peptidergic neuron. In *unc-39* mutants, there is failure of RID neurogenesis. The absence of a neurite with neuroendocrine properties in the dorsal nerve cord of *unc-39* mutants confirms that the one that we observed in the wild-type dorsal nerve cord was RID. 2) RID promotes forward run. The motor behaviors of *unc-39* mutant reiterate those that we observed for RID-ablated animals; the absence of a behavioral response to optogenetic stimulation of the RID neuron serves as a control and confirms the cellular origin of the effect was RID. 3) FLP-14 and other neuropeptides are expressed by RID. Since we do not have a RID-specific promoter, we took advantage of *unc-39*’s absence of RID to perform subtractive RNA profiling to search for RID-enriched transcripts. Indeed UNC-39 transcriptional targets are likely present in this list, but this was not our focus. The sole reason to provide those data was to be useful to those who are interested in pursuing UNC-39’s targets, and we are confident of the reproducibility and quality of these datasets.

This is why our *unc-39*-relatedresult sections focused on evidence for RID’s absence in *unc-39* mutants; the rest of the *unc-39* work was distributed in our Results sections where they are integrated as controls. In the Discussion, we now further highlight the relevance of the *unc-39* data to the overall theme of the paper: the absence of RID in *unc-39* mutants has provided our work a genetic tool to investigate the property, function, and molecular mechanisms of RID. We further refer to published work on *Drosophila* Six; together they suggest that UNC-39/Six/SO transcription factors may have a shared role in endocrine development.

*During forward movement, calcium levels in RID increased, whereas forward to reverse transitions caused a decrease in calcium levels, suggesting that RID is involved in forward movement. The calcium imaging was supported by behavioral work that showed that unc-39 mutants and RID-ablated animals had shorter forward runs and more turns than wild type, although unc-39 mutants had a more severe phenotype than the RID-ablated animals (Figure 1), suggesting that unc-39 regulates other neurons that sustain forward movement. The authors followed up by showing that optogenetic stimulation of RID caused a sustained increase in forward velocity and runs.*

*The authors performed a subtractive transcriptome analysis to identify three candidate neuropeptide genes that are enriched in RID: flp-14, ins-17, and nlp-34. flp-14 mutants show a decreased average forward run duration, an increased number forward initiations, and an increased number of reversal initiations compared to wild type. However, as the authors note, the defects in flp-14 are not as severe as in unc-39 mutants and RID-ablated animals, suggesting that other peptides may act in concert with FLP-14 to modulate forward movement.*

**Response:** These are our conclusions. We agree that behavioral differences summarized by bar graphs may not be sufficiently informative for readers without direct experience performing *C. elegans* motor behavior assays. To place these quantitative analyses in a more visual context, we now provide supplemental figures where we show the raw data that were used for quantification. They allow a qualitative, but very intuitive view of the data variability of the same genotype, and the degree of difference across groups. It should be evident from these raw data that the motor phenotypes of *unc-39* and RID-ablated animals were quite similar, and *flp-14*’s defect was of the same trend but less severe compared to the *unc-39* or RID-ablated animals. These points have been articulated in Results and Discussion.

*In addition, the behavioral assays were highly variable (i.e., the SEMs are very large, suggesting a lot of variability in the behaviors). For instance, the wild-type values for average forward duration were significantly different in Figure 5. This variability makes looking at defects in flp-14 mutants and its rescue difficult; for instance, 1) the average forward duration for flp-14 mutants looked significantly different from wild type in Figure 5, but not really in Figure 5; 2) the flp-14 mutants were not rescued with a copy of the flp-14 transgene for average forward duration to levels seen in wild type or even the flp-14 transgenics alone (Figure 5); and 3) the flp-14 mutants did not look significantly different from the flp-14 transgenic line (Figure 5). The only consistent defect seen with the flp-14 mutants was the decrease in the number of forward initiations. The authors should re-word some of their statements.*

**Response:** We appreciate the reviewer’s keen observations on the behavioral defects of *flp-14* mutants.

a) We now provide raw behavioral recording data (Figure 5—figure supplement 1 and Figure 6—figure supplement 1 and Figure 6—figure supplement 2) so readers can view directly the variability between strains;

b) Regarding the concern about the variability, and *flp-14* rescue in former Figure 5 was not back to the wild-type level in another panel (former Figure 5), we need to clarify several points. First, ‘wild-type’ animals in the original 5A and 5B/C refer to animals of different genetic backgrounds, because they served as the wild-type control for each experimental group. In Figure 5, ‘wild-type’ was N2, a non-transgenic strain where *flp-14* and *unc-39* mutants were derived from. In Figure 5, ‘wild-type’ was *Si(FLP-14)*, a transgenic line with a single copy integrant of the FLP-14 rescuing construct. In Figure 5, ‘wild-type’ was *Si(FLP-14);hpIs202* – *hpIs202* being a GFP marker for visualizing RID. These controls were not to be compared with each other. The genotype information is provided in figure legends. Second, even for strains of the identical genetic background, the absolute value of the motor behavioral index would exhibit variability when they are assayed on different days. In our behavioral assays, we only compare values of animals assayed on the same day and on the same plate, as we explained in the Materials and methods and Appendix 1. The variability we observed is standard for these motor behavioral assays that were performed by our group, so we are confident with conclusions drawn from these data.

c) Regarding the concern that *Si(FLP-14)* did not rescue *flp-14* back to *Si(FLP-14).* There was indeed one motor index between *flp-14;Si(FLP-14)* and *Si(FLP-14)* that showed really small, but statistically significant difference. An exogenous or even an overexpression of neuropeptides could easily generate effects on motor behaviors, and it was difficult to assert cellular origin of secreted molecules by standard transgenic rescue experiments. We generated a single copy insert of *flp-14* construct *Si(FLP-14)* under a portion of its own genomic regulatory elements to avoid these caveats as much as possible, but it is quite possible that *flp-14;Si(FLP-14) did not behave identical to Si(FLP-14)*. With the raw data, the rescuing effect should be quite obvious. The key point of this experiment was to ablate the RID neurons in the *Si(FLP-4)* background to determine if its rescuing effect was reversed.

We have reworded the statement to reflect these points.

*The authors indicate that ins-17 mutants acted similarly as wild typ￼e and the double mutants acted similar to the flp-14 mutants alone. However, flp-14 ins-17 double mutants showed more severe phenotypes than flp-14 mutants alone (Figure 5—figure supplement 1), contrary to what the authors indicate, suggesting that INS-17 peptides might also be contributing to forward movement. Their statement should be re-worded.*

**Response:** In the original Figure 5—figure supplement 1, *flp-14 ins-17* double mutants were compared against wild-type and *ins-17*. During revision, we realized that we had forgotten to replace an older version of the figure, resulting in the absence of the *flp-14* panel in the previous submission altogether.

We have replaced it with the proper panel. We moved these results to the main figure and provided raw behavior data as a supplemental figure. As shown in Figure 5 and supplemental raw data (Figure 5—figure supplement 2) in the resubmission, *ins-17* did not significantly enhance *flp-14*’s motor defects. Double mutants did exhibit a trend of slightly decreased reversal duration, forward, and reversal velocity when compared to *flp-14* alone, but the change is so small, excluding *ins-17* being a main regulator of motor behaviors. We have reworded the text to state that *ins-17* is not a main contributor. We thank the reviewer for bringing it to our attention.

*Analysis of the possible FLP-14 receptors was not very informative. Double knockout of the possible receptors, NPR-4 and NPR-11, showed phenotypes that were not significantly different than wild type, but showed the same general trend as flp-14 mutants. Despite this negative result, Figure 7 should be included in the text rather than in the Discussion.*

**Response:** Agree. We have incorporated this data into the Results section (see Figure 8 in resubmission).

*The paper lacks a model of how RID inputs into the motor circuit to sustain forward motion. RID synapses onto hypodermal cells, DD neurons, and muscle, but could also act humorally. Do the authors hypothesize that the effect of FLP-14 peptides released from RID is on DD neurons, muscle, or other targets? In Ascaris, application of FLP-14 on muscle causes relaxation followed by rhythmic contractions or sustained contractions (depending on which paper one reads). The authors should have a model for the neuromodulation.*

**Response:** Indeed. We do not have any positive data or conclusions on RID’s downstream targets. As we previously discussed, the *Ascaris* electrophysiology work alluded to a possibility of muscles and forward driving motor neurons being the potential targets. This notion would contradict the candidate neurons based on expression pattern of the predicted FLP-14 receptors (interneurons). Because of our negative data on these receptor mutants, we favor clues from the *Ascaris* work.

To make a speculative model that does not mislead other researchers, we placed question marks on these candidate targets in the model sketch. In the Discussion, we further emphasize the necessity to identify FLP-14’s physiological receptors in order to determine RID target cells. Although DD and hypodermis were the postsynaptic partner of RID in the adult wiring diagram, this was based on their anatomic apposition to the few active zones observed along the RID axon. This approach may have its limitations because ~90% of RID axonal boutons lack active zones; for the ~10% with active zones, we never observed docked clear or dense core vesicles.

Reviewer #2:

*1) Specificity of the activation experiments with Chrimson. Please explain more in detail how the authors restricted the expression of Chrimson to RID neurons using the ubiquitin ligase system. Since the specificity of Chrimson expression is critical for the interpretation of the results, inclusion of a picture showing the specific expression as a supplementary figure would be desirable.*

**Response:** We added the description in the Results, and expanded the methods (see Appendix 1) on obtaining these strains. A supplementary figure containing the requested image has been added (Figure 7—figure supplement 1).

*I also wonder how one should interpret the results of the optogenetic activation in unc-39 mutants. Is the reversion of the phenotype due to lack of Chrimson expression in RID or defective function of the neuron?*

**Response:** Because RID neurogenesis fails in *unc-39* mutants, the lack of Chrimson expression in RID and subsequently the absence of an activation response to light stimulation serves as a negative control to ensure that the effect of optogenetic light stimulation was dependent on the presence of the RID.

*2) It is not clear how RID regulates forward locomotion. I would expect more through discussion on putative target cells of RID in relation to known locomotory circuits. An obvious candidate would be DD MNs whose terminals reside in the immediate neighbor of RID varicosities. What happens to forward locomotion when DD MNs are inhibited or activated? If feasible, the authors may also examine if the activity manipulation of RID or application of FLP-14 affects the level of Ca^2+^ or other signaling molecules (e.g., cAMP) in DDs.*

**Response:** We thank the reviewer for this suggestion. We do not understand how RID regulates forward locomotion in the context of other circuit components. We provide more background and expand Discussion in the revised manuscript. DD could be a candidate due to its proximity, but other motor neuron neurites are also fairly close. A main issue is that the post-synaptic partner of peptidergic neurons could not be reliably predicted by anatomic apposition. We also suspect that the predicted FLP-14 receptors are incorrect: we did not find any supportive experimental evidence for the predicted FLP-14 receptor mutants (Figure 8 in Resubmission). The proposed experiments will be informative, but we feel that it is only the starting point for deciphering the RID circuit. It should be the focus of future investigation.

Reviewer #3:

*1) There seems to be a caveat implied by Figure 6 that should be discussed. The speed of unstimulated flp-14 mutants was significantly higher than that of wild-type while the stimulated speeds of the two strains were much more similar. Therefore, a possible (plausible?) interpretation of the difference in fold change is a ceiling effect. The animals, on average, didn't move faster than ~0.55 body-lengths per sec such that the unstimulated mutants didn't have as much capacity for speedup as wild-type. The data in panel C possibly suffers from a similar (not unrelated) issue: T_off for the mutants is higher than wild-type.*

*If so, the optogenetic assay does not strongly support the conclusions of the manuscript although it doesn't contradict them either. It is perfectly reasonable for a particular assay of a complex phenomenon to fall short of providing conclusive evidence. It is even useful to present such results by way of transparency and completeness. Thus, the data is useful and should be presented in the main text without detracting from the interest of the manuscript or the validity of the conclusions.*

*If I am mistaken, perhaps the caveat should be briefly mentioned and refuted for the sake of clarity. Otherwise, the caveat should be discussed and the conclusion of the relevant Results section should be revised to state that the optogenetic assays were consistent with the overall theme of the work but in and of themselves inconclusive.*

**Response:** We thank the reviewer for bringing this observation to our attention. We have re-examined all data to see if there is a general trend of higher activity in *flp-14* mutants before stimulation when compared to other animals, and whether the post-stimulation velocity represented the highest velocity in these recordings. This was not the case, and we take this opportunity to elaborate on these experiments and analyses done.

In Figure 7 (previously Figure 6), we had purposely avoided comparing the mean velocity across strains because it would not be accurate. For a velocity comparison between wild-type and *flp-14*, we refer to Figure 5. These assays, where we compared animals of matched age on the same day and on the same plate, provide the accurate assessment. As shown in Results, there was no substantial difference between the mean spontaneous forward velocity between wild-type and *flp-14*.

To reiterate the methodology used for the optogenetic stimulation experiment (we described it in detail in Appendix 1, shortened here), we recorded one animal at a time, with a 3 minute light on/3 minute light off protocol for 30 minutes, or until we lost track of the animal before 30 minutes. Stimulation on and off events were pooled (N) from recordings of the same genotype, and we compared the motor behaviors with and without stimulation within the same recording using two indexes: a) The mean duration of all forward run events during the ON and OFF periods (Figure 7; previously Figure 6). b) The velocity before and after the simulation (Figure 7; previously Figure 6). The index from 7A is directly relevant to our conclusion – RID activation positively correlates with prolonged forward run. We were glad we also saw a trend of increased velocity after RID stimulation, which supported, but was not definitive for our conclusion.

The reviewers may have noticed that the N number for quantified velocity change before and after stimulation (Figure 7) is much smaller than the N number for the distribution graph of forward duration for ON and OFF period (Figure 7). This was because the motor state of the animal, when the light was on, was random. There was a concern with a timing effort (e.g. the animal was at a low or high velocity forward state when the stimulation was turned on), or, false negative effect (e.g. we hit the animal when it was reversing). Hence we selected for stimulation events that happened within a forward run, with at least 4 seconds before and after the stimulation. The previous choice of 4s was set by the motor defects of *unc-39* mutants: these animals almost exclusively exhibited short runs; if we chose a longer time, N from the *unc-39* dataset would be too small.

These experiments were very labor-intensive. Because only 10% of our transgenic animals exhibited removal of Chrimson from non-RID neurons (see Appendix 1 for details), we picked as many candidate RID-specific old larvae to young adult animals as possible from three genotypes under low magnification fluorescent microscopes, cultured them on retinal plates, and recorded all animals on the following day. This amounted to 12-14hr recording, for three genotypes of ~6~10 animals/genotype). After recording, we mounted each animal to examine with a high magnification microscope, and discarded data from animals that showed residual chrimson signals in neurons other than RID. Three independent repeats were performed. The scarcity of samples prevented us from using animals at the identical developmental stage, and we had to pool recording data obtained on different dates. Together with the small number of Ns for velocity comparison, we felt that we should only use these datasets to compare the motor behavior within the same recording, hence the normalized difference at the ON and OFF period. In fact, we knew that the absolute velocity value in these panels did not represent the spontaneous velocity as observed under culturing conditions – *unc-39* exhibited a non-ambiguous drastic reduction in velocity when compared to both wild-type and *flp-14* animals (Figure 5 and videos), which was not reflected by the original Figure 6 (Figure 7—figure supplement 1) at all.

In revision, we place the more relevant, run duration panel ahead of the velocity panel. We also present the velocity values using scatter plots for an easier comparison between individual pairs. Moreover, we have now examined and presented the mean velocity before and after stimulation for 10 seconds for wild-type, *flp-14* and *npr-4 npr-11* mutants (a more representative mean speed), while keeping *unc-39’s* cut off to 4 seconds. These data led to the same conclusion (Figure 7 and Figure 8).

*2) Figure 7 depicts a negative result, suggesting that the predicted receptors of FLP-14 do not directly affect forward locomotion (either because the prediction is wrong or because their interaction with multiple ligands is more complex). I think that such data does belong in a Results section. It is of practical importance to C. elegans researchers and of conceptual importance to any researcher that felt the need to tuck away a valid result in the name of a 'cleaner', more linear story.*

**Response:** Agreed. We moved Figure 7 (Figure 8 in the resubmission) to the main Results section.